# MindCraft: How Concept Trees Take Shape In Deep Models

## Abstract

Large-scale foundation models demonstrate strong performance on language, vision, and reasoning tasks. However, how they internally structure and stabilize concepts remains elusive. Inspired by causal inference, we introduce the **MindCraft** framework built upon **Concept Trees**. By applying spectral decomposition at each layer and linking principal directions into branching Concept Paths, Concept Trees reconstruct the hierarchical emergence of concepts, revealing exactly when they diverge from shared representations into linearly separable subspaces. Empirical evaluations across diverse scenarios across disciplines, including medical diagnosis, physics reasoning, and political decision-making, show that Concept Trees recover semantic hierarchies, disentangle latent concepts, and can be widely applied across multiple domains. The Concept Tree establishes a widely applicable and powerful framework that enables in-depth analysis of conceptual representations in deep models, marking a significant step forward in the foundation of interpretable AI.

## 1 Introduction

Deep learning has achieved remarkable success across diverse domains, including computer vision (Krizhevsky et al., 2012), text generation (Devlin et al., 2019), and speech recognition (Hinton et al., 2012; Graves et al., 2013). However, the internal mechanisms of neural networks remain opaque. Despite rapid progress in visualization and interpretability techniques, we still lack a clear understanding of how abstract concepts, such as "*disease*," "*cause*," or "*truth*", form and stabilize inside their layers (Lipton, 2016; Doshi-Velez & Kim, 2017; Ribeiro et al., 2016). This opacity has fueled the widespread description of neural networks as "black boxes" Lipton (2016); Doshi-Velez & Kim (2017), raising concerns about reliability in sensitive domains such as healthcare (Caruana et al., 2015), finance (Rudin, 2019), and law Doshi-Velez & Kim (2017).

Recent advances in *representation analysis* have opened promising avenues for transparency. For example, networks trained to play chess acquired a range of human chess concepts (McGrath et al., 2022). Similarly, generative and self-supervised models exhibit emergent representations such as semantic segmentation in vision tasks (Caron et al., 2021; Oquab et al., 2023). Zou et al. (2023) formalized this line of work as *Representation Engineering (RepE)*, showing how we can extract concept directions from a model's internal states and even control its behavior. RepE provides a *top-down* view of interpretability, shifting focus from individual neurons or circuits to global representational structure. In parallel, the *Linear Representation Hypothesis (LRH)* (Park et al., 2023) suggests that task-relevant concepts gradually become linearly separable while the input propagates through deeper layers of the model, enabling simple linear probes and edits to access them.

However, both perspectives remain limited. RepE reveals *where* concepts can be extracted but not *how* they emerge or why they stabilize. LRH describes separability but not the dynamics that give rise to it. In practice, these approaches overlook the process **by which local perturbations propagate through layers and self-organize into stable conceptual hierarchies**. This gap poses a fundamental question unanswered: *How do neural networks internally construct and consolidate abstract concepts?*

We address this question with **MindCraft**, a novel framework that traces counterfactual difference propagation, which is the process of following how a small, targeted change in the input alters internal representations as it moves through the layers of the network. For example, flipping "*the

*patient has diabetes*" to "*the patient has hypertension*", creates a counterfactual pair. By tracking how the resulting difference vector evolves across layers, we can identify branching points where concepts diverge into distinct and stable subspaces.

Understanding this process is not just an academic exercise. It enables precise debugging (by locating where models confuse concepts), fairness auditing (by exposing where sensitive attributes split), and accountability in high-stakes domains. More broadly, MindCraft reframes interpretability: rather than treating models as static black boxes, it reveals the layer-wise dynamics by which abstract reasoning structures are created and maintained. Beyond theoretical insight, MindCraft also provides a practical tool. We design an automated pipeline that takes arbitrary text as input and produces Concept Trees on demand. This enables scalable analysis across domains and lowers the barrier for applying interpretability methods in practice.

Our main contributions are as follows:

- To our knowledge, **MindCraft** is the first method that systematically traces ***how*** abstract concepts emerge, branch, and stabilize across layers, moving beyond prior work that only probes static representations.

- We introduce a counterfactual propagation protocol combined with spectral decomposition of attention value projections, yielding sensitive and robust measures of conceptual separation.

- We apply MindCraft to extensive reasoning scenarios, including medical diagnosis, physics reasoning, and political decision-making, demonstrating its ability to recover semantic hierarchies, disentangle latent concepts, and generalize across tasks.

- We provide an automated pipeline that enables scalable, on-demand analysis of conceptual representations, advancing the foundations of transparent and trustworthy AI.

## 2 RELATED WORK

### 2.1 REPRESENTATION LEARNING IN DEEP NETWORKS

Representation Learning studies how neural networks develop internal representations that encode task-relevant information. Early research on word embeddings demonstrated that neural networks can capture rich semantic and syntactic relationships in a distributed manner (Mikolov et al., 2013). Subsequent studies showed that such representations often reflect abstract latent factors: for example, Radford et al. (2018) observed the emergence of sentiment-tracking units, and Schramowski et al. (2019) reported that large language models (LLMs) encode implicit moral dimensions.

Similar phenomena have been observed beyond language. In reinforcement learning, McGrath et al. (2022) found that models trained to play chess develop internal abstractions of board state and strategy. In computer vision, self-supervised and generative objectives have been shown to induce semantic feature maps useful for downstream tasks such as segmentation (Caron et al., 2021; Oquab et al., 2023). Building on these observations, Zou et al. (2023) introduced *Representation Engineering (RepE)*, which uses contrastive edits to extract and manipulate concept directions from internal activations. Their work exemplifies a top-down approach to interpretability: instead of focusing on single neurons or weights, it analyzes the global structure of representations. Similarly, Park et al. (2023) proposed the *Linear Representation Hypothesis (LRH)*, arguing that task-relevant concepts become linearly separable with depth, enabling simple linear probes and edits.

### 2.2 NEURAL NETWORK INTERPRETABILITY

Many interpretability methods emphasize parameter or gradient-based structures, seeking to explain model behavior in terms of weights, activations, or neuron circuits. Saliency-based methods including Simonyan et al. (2013); Springenberg et al. (2014); Zhou et al. (2016) highlight influential input regions via gradients or activations, while feature visualization (Zeiler & Fergus, 2014) synthesizes inputs that strongly activate particular neurons. More recently, mechanistic interpretability aims to reverse engineer networks into circuits of interpretable components (Olah et al., 2020; Olsson et al., 2022; Lieberum et al., 2023). Although powerful, these approaches can be brittle or require

extensive manual effort, and they typically do not explain how abstract representations form from distributed activations.

Recent works instead analyze the *geometry and dynamics* of representation spaces. Techniques such as CKA (Kornblith et al., 2019), SVCCA (Raghu et al., 2017), and Procrustes alignment (Gao et al., 2021) measure how representations evolve across layers or between models. Other studies have investigated how residual connections promote stable feature reuse and facilitate optimization (He et al., 2016; Dehghani et al., 2023), and how singular value spectra reflect information propagation and compression in deep networks (Saxe et al., 2014; Morcos et al., 2018; Canatar et al., 2021). These lines of work suggest that deep networks may progressively concentrate representational energy along a few dominant directions, a phenomenon that we directly exploit.

## 3 PRELIMINARIES

### 3.1 COUNTERFACTUALS

We adopt the notion of *counterfactual* from causal inference (Pearl, 2009), defined as the resulting variable after an *intervention* is applied to a particular variable (e.g., a token), while the surrounding *context* is held constant. The resulting variable, observed after the intervention, is the counterfactual to the original variable. Specifically, if we intervene on a specific component $x$ of $X$, and calculate the counterfactual on the general target variable $Y$, we write:

$$Y_{x \leftarrow \Delta x} \;=\; Y \mid (do(x = \Delta x), X \setminus x), \tag{1}$$

where $do(\cdot)$ denotes the intervention that assigns $x$ the value $\Delta x$, and $X \setminus x$ means the remaining context that is constant through the intervention. For example, consider sentiment classification with input sequence: "The movie was great." The model predicts $Y = $ Positive. We intervene on the token $x = $ "great" by replacing it with $\Delta x = $ "terrible": "The movie was terrible.", with other tokens $X \setminus x$ remaining the same, yielding the counterfactual $Y_{x \leftarrow \Delta x} = $ Negative. For notational convenience, the counterfactual $Y_{x \leftarrow \Delta x}$ can be simplified as $Y_{\Delta x}$. The difference between the counterfactual and the original is:

$$\delta Y \;=\; Y_{\Delta x} - Y, \tag{2}$$

where we call $\delta Y$ the counterfactual difference in $Y$, and $(Y_{\Delta x}, Y)$ is the counterfactual pair. Specifically, the Concept Tree focuses on observing the counterfactual pair of the last token in the user-provided input sequence, which we will specify in the methodology.

## 4 METHODOLOGY

### 4.1 MOTIVATION: PROPAGATION OF COUNTERFACTUAL SIGNALS

Our starting point is the simple idea that a model's understanding of concepts can be revealed by observing how it processes counterfactual edits that are small targeted input changes. We begin by performing an intervention on the input sequence:

```
X:      You are a powerful  leader making decisions.
X_Δx:  You are a powerless leader making decisions.
```

The only difference is the word change (powerful → powerless). By analyzing how this minimal intervention propagates through the network (see Figure 1), we can observe where the model begins to treat the two inputs differently. Following prior work showing that the last token often best captures a model's generative state (Meng et al., 2022; Zou et al., 2023), we focus on the value representations of the final token in the sequence, denoted as $V_{L(\Delta x)}^{(-1)}$ and $V_L^{(-1)}$, where the superscript $-1$ indicates the last token in the sequence. Comparing these across layers reveals how the counterfactual difference evolves.

Figure 2 shows the cosine similarity between the factual and counterfactual representations at each layer. In the early layers (roughly before layer 10), similarity remains high, meaning the network has not yet distinguished the two contexts. In mid-layers, however, similarity drops sharply (red bars), indicating a sudden amplification of the conceptual difference. Beyond these branching points, the

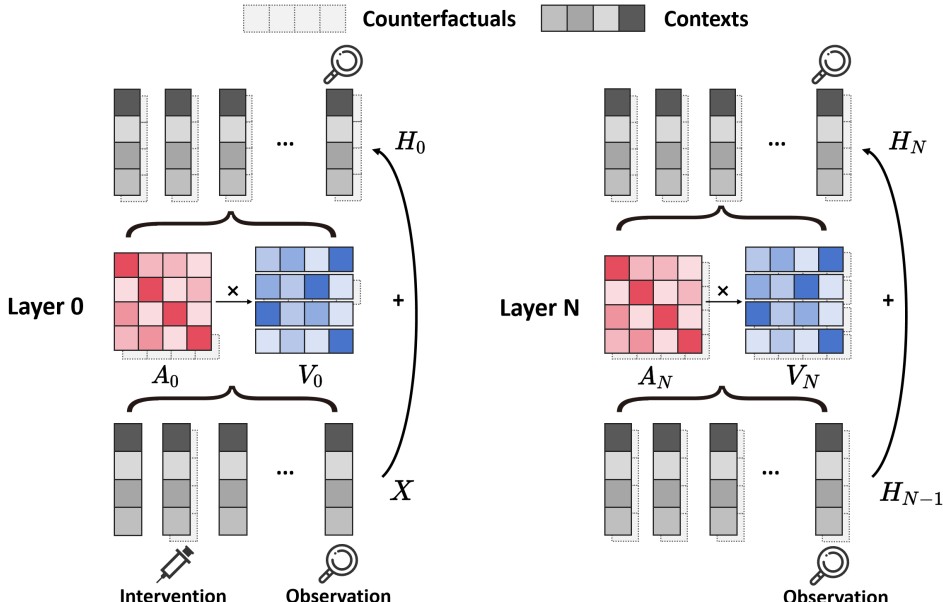

Figure 1: Overview of the MindCraft algorithm, where $X$ is the input sequence, for layer $N$, $H_N$ is the attention output sequence after residual connection, $A_N$ is the attention weight, $V_N$ is the value matrix. We first perform an intervention on a specific input token. Then, leveraging the attention mechanism, we compare between the counterfactual and the original representation at the last token. This difference reveals the hierarchical layer at which concepts separate within the model.

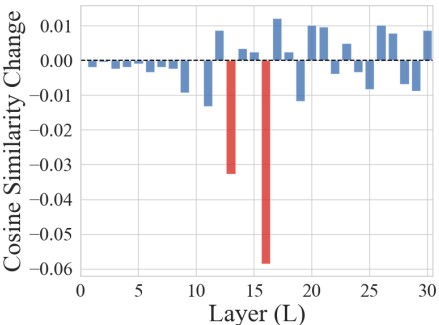

Figure 2: $\cos(V^{(-1)}_{L(\Delta x)}, V^{(-1)}_{L})$ change, where a sudden amplification of the conceptual difference is observed.

representations stabilize again, converging toward a consistent trajectory in deeper layers.

This observation suggests that the model distinguishes between "*powerful*" and "*powerless*" precisely at these layers, where counterfactual differences first become salient. Concept formation, therefore, follows a **branch-and-stabilize process**: representations remain similar in early layers, diverge sharply at branching points, and then stabilize into distinct subspaces. Such a process highlights that concept-level organization is not static, but unfolds progressively through the network. This dynamic motivates our central abstraction, the **Concept Tree**, which models concept emergence as a hierarchical process of extraction, divergence, and stabilization across layers.

### 4.2 PAVING THE CONCEPT PATH

The above observations suggest that counterfactual differences do not diffuse randomly through the network. Instead, they follow structured routes: *remaining latent in early layers, sharply diverging at branching points, and then stabilizing into consistent directions*. To formalize this process, we introduce the notion of a **Concept Path**, a representation of how a counterfactual signal propagates through the network's layers.

To do so, we begin with the self-attention mechanism, which computes three matrices from the input representations of all tokens. Given a sequence of input token representations from the previous layer, $Z \in \mathbb{R}^{n \times d}$, where $n$ is the sequence length and $d$ is the model dimension, the mechanism computes Query ($Q$), Key ($K$), and Value ($V$) matrices:

$$Q = ZW_Q, \quad K = ZW_K, \quad V = ZW_V \qquad (3)$$

where $W_Q, W_K, W_V \in \mathbb{R}^{d \times d}$ are learnable weight matrices. The output of the attention head is a weighted sum of the Value vectors:

$$\text{Attention}(Q, K, V) = \text{softmax}\left(\frac{QK^\top}{\sqrt{d_k}}\right) V \tag{4}$$

where $d_k$ is the dimension of the keys.

Our analysis focuses on the last token in the sequence, since it integrates information from the full context and directly conditions the model's next prediction (Meng et al., 2022; Zou et al., 2023). By examining how this representation changes across layers, we can trace the emergence of conceptual differences. Therefore, we focus on the last-token representation, denoted $Z^{(-1)} \in \mathbb{R}^d$, as the anchor point for our spectral analysis. While the Value vector of the last token already carries rich contextual information, directly analyzing it can be noisy and unstable. To obtain a more robust basis, we examine the Value transformation matrix $W_V$ using singular value decomposition (SVD):

$$W_V = U \Sigma R^\top \tag{5}$$

where $U\Sigma R^\top$ is its SVD with $U = [u_1, \ldots, u_m]$, $\Sigma = \text{diag}(\sigma_1, \ldots, \sigma_p)$, $R = [r_1, \ldots, r_n]$, and $p = \min(m, n)$. These left singular vectors, $u_i$, represent the principal directions along which the transformation $W_V$ has the most significant amplifying or attenuating effect, as determined by the corresponding singular values in $\Sigma$. This basis provides a more stable and meaningful coordinate system to analyze than the content of the raw Value vector, which is further testified in Appendix B.1. With this setup, we define the **Concept Path** as the decomposition of the last token's Value vector across all principal directions of $W_V$. This gives us a spectral signature of the token's information content.

**Definition 1** (**Concept Path**). *For a self-attention layer, the **Concept Path** $\mathcal{C}$ for the last token is the vector of projections of its Value vector, $v = V^{(-1)}$, onto the complete basis of left singular vectors, $\{u_i\}_{i=1}^d$, of the matrix $W_V$.*

$$\mathcal{C} = \left[ \langle v, u_1 \rangle \sigma_1, \ \langle v, u_2 \rangle \sigma_2, \ \ldots, \ \langle v, u_p \rangle \sigma_p \right] \in \mathbb{R}^p. \tag{6}$$

Essentially, the Concept Path vector $\mathcal{C}$ is the representation of $V^{(-1)}$ in the coordinate system defined by the principal axes of the Value transformation. Each component quantifies how much of the last token's semantic content is aligned with the $i$-th principal direction. By tracking how this spectral signature $\mathcal{C}$ evolves across layers, we can precisely trace the dynamics of concept formation. Following this path, we therefore organize it into the Concept Tree, where nodes correspond to shared spectral directions and edges reflect how distinct concepts separate as they propagate through the deep neural network.

## 4.3 Constructing the Concept Tree

The Concept Path provides a spectral signature of a token's representation at each layer. To understand how a model gradually distinguishes between different concepts, we extend this idea into the **Concept Tree**, a structure that captures where concepts separate.

Consider an original input $X$ and its counterfactual version $X_{\Delta x}$. For each layer $l$ in the network, we compute their last-token Concept Paths, denoted by:

$$\mathcal{C}_l(X) \quad \text{and} \quad \mathcal{C}_l(X_{\Delta x}) \in \mathbb{R}^p \tag{7}$$

These vectors summarize the decomposition of the token's Value representation along the principal directions of the Value projection.

Because the singular value decomposition (SVD) orders these directions by importance, most of the semantic content is concentrated in the top few components. To reduce noise and highlight the core signal, we filter each Concept Path by retaining only its top-$k$ components. Specifically, let $\text{topk}(\mathcal{C}, k)$ be an operator that takes a vector $\mathcal{C}$ and an integer $k$, and returns a new vector where only the $k$ components of $\mathcal{C}$ with the largest absolute values are preserved, and all other components are set to zero, and therefore we obtain:

$$\tilde{\mathcal{C}}_l(X) = \text{topk}(\mathcal{C}_l(X), k) \tag{8}$$

$$\tilde{\mathcal{C}}_l(X_{\Delta x}) = \text{topk}(\mathcal{C}_l(X_{\Delta x}), k) \tag{9}$$

To measure how similarly the model treats the two inputs at layer $l$, we compute the **Conceptual Separation Score**, $s_l$, as the cosine similarity between the two filtered Concept Paths:

$$s_l(X, X_{\Delta x}) = \cos\left(\tilde{\mathcal{C}}_l(X), \tilde{\mathcal{C}}_l(X_{\Delta x})\right) = \frac{\tilde{\mathcal{C}}_l(X) \cdot \tilde{\mathcal{C}}_l(X_{\Delta x})}{\|\tilde{\mathcal{C}}_l(X)\|\|\tilde{\mathcal{C}}_l(X_{\Delta x})\|} \tag{10}$$

A score close to 1 indicates that the model still treats the inputs similarly, while lower scores indicate that the representations are diverging.

Formally, we define the **Branching Layer** $l^*$ as the first layer where the similarity drops below a threshold $\tau$ (e.g., $\tau = 0.9$):

$$l^*(X, X_{\Delta x}) = \min\{l \in [0, L-1] \mid s_l(X, X_{\Delta x}) < \tau\}. \tag{11}$$

This marks the point at which the model begins to robustly separate the concepts. If the score never falls below $\tau$, the concepts are considered inseparable by the model. With this machinery, we provide a formal, bottom-up definition of the Concept Tree.

**Definition 2** (**Concept Tree**). *A **Concept Tree** $\mathcal{T}$ is a hierarchical structure that visualizes the separation layers for a set of input concepts. The root of the tree represents all concepts being undifferentiated at layer 0. A branch emerges from a parent node at layer $l$ if $l$ is the Separation Layer $l^*$ for a subset of the concepts within that node. Each node in the tree at depth $l$ corresponds to a cluster of concepts that have not yet been separated from each other up to layer $l - 1$.*

In practice, the tree is constructed by analyzing the Separation Layer $l^*$ for all relevant pairs of counterfactual inputs. The distribution of these $l^*$ values reveals the model's decision-making hierarchy: early branches correspond to coarse-grained distinctions, while later branches signify finer-grained semantic processing.

## 5 EXPERIMENTS

### 5.1 CONCEPT TREE IMPLEMENTATIONS

The experiment results from a variety of scenarios shown in Figure 3. For concept identification, we incorporate the automated pipeline as demonstrated in Appendix C. We use $k = 10$ and $\tau = 0.9$ by default, and more details about experiment settings are in Appendix D. The results demonstrate the broad applicability of the concept tree, and we find that the model's internal processing of concepts is indeed not a monolithic, linear process but rather a structured, hierarchical tree-shaped organization. This tree serves as a powerful visualization, mapping the pathway of concept transformations—from the most influential conceptual directions to progressively less contributive ones—that the model follows to derive a conclusion, such as reasoning from one fact to another.

For instance, in the medical diagnosis case (a), the Concept Tree reveals that treatment- and symptom-related distinctions such as "metformin/insulin" and "diabetes/hypertension" branch earlier and carry greater importance than temporal variations like "March/July." In the daily-life scenario (c), concepts that determine the sentiment or tone of the sentence, such as "mom/dad" and "good/bad," dominate the branching structure, while numerical differences like "99/100" are treated as less critical. Similarly, in the physics reasoning example (e), fundamental notions such as "physics" and "break" emerge as primary branching factors, whereas objects like "building/table" appear later and play a more minor role. These examples demonstrate that the Concept Tree highlights conceptually salient distinctions rather than surface-level token differences, offering a structured account and board application of how models prioritize concepts across layers.

### 5.2 CONCEPT HIGHLIGHT EXPERIMENT

To further validate that the Concept Tree framework captures semantic interpretability rather than relying solely on static measures such as input or latent embeddings, we perform an experiment in Figure 4. In case (h), the model processes a standard sentence without explicit emphasis, where temporal tokens such as "2024" and "2025" behave similarly to other factual tokens. In contrast,

**(a)** The patient was screened positive with type 2 diabetes in March 2023, and is currently taking metformin twice daily. Based on these findings, provide the most suitable treatment:

**(b)** The man has a cut on the arm and it is bleeding. Based on these findings, give a treatment plan:

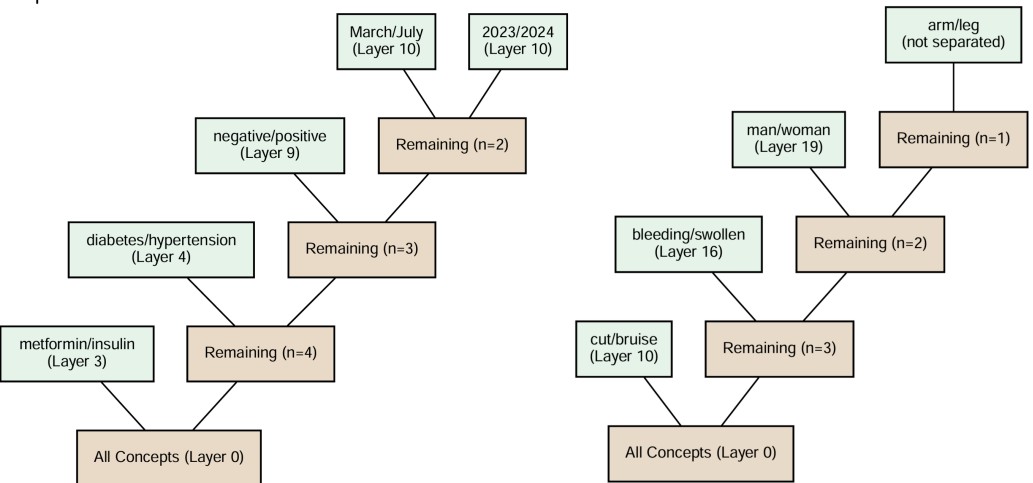

**(c)** I got a good grade on my exam — 99. Today is January 1st, 2025. My mom said that because of this, I should be:

**(d)** I always help my classmates with their homework and listen to their problems. My friends say this means I am kind anyway:

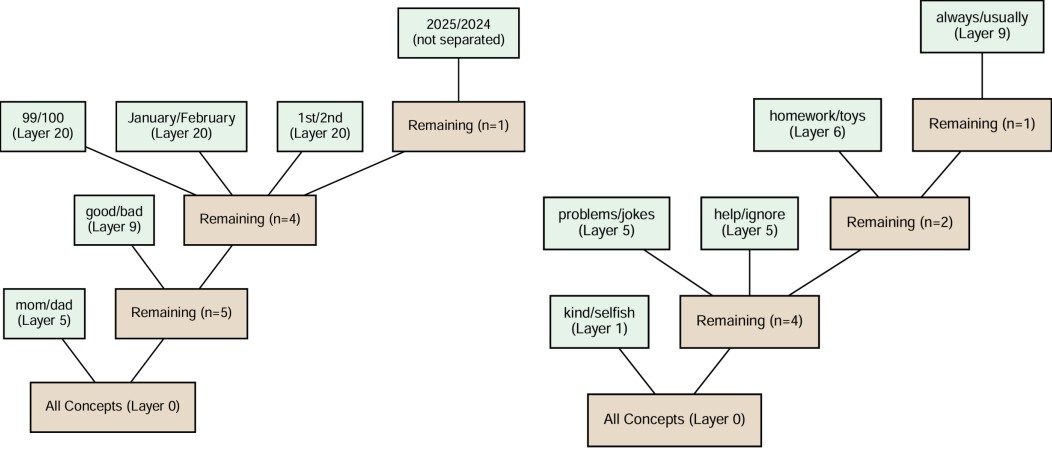

**(e)** I dropped a glass ball from the top of a building, and it hit the ground. According to physics, what will happen to the ball after it hits the ground? Break or not?

**(f)** The city mayor decided to make bus rides free for everyone. How will most people in the city probably feel happy about this decision?

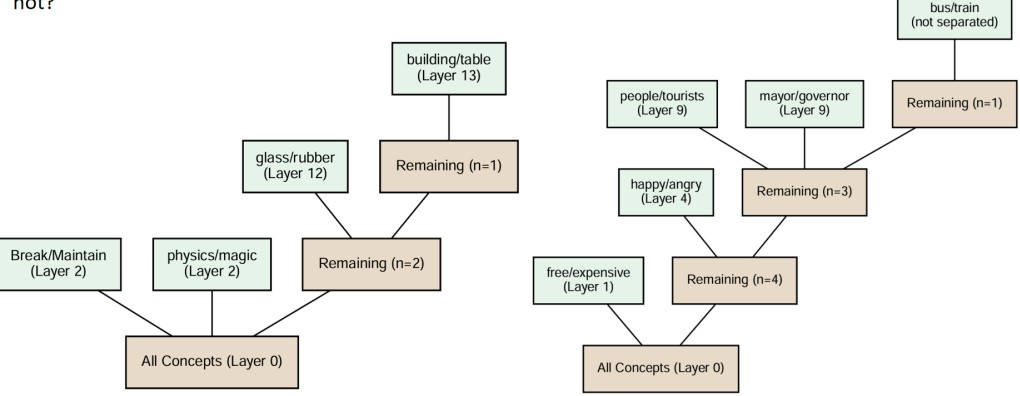

Figure 3: Concept Trees constructed from six scenarios: (a–b) medical diagnosis, (c) daily life, (d) personality evaluation, (e) physics reasoning, and (f) political decision-making. Counterfactual tokens are marked with underlines, and n denotes the number of remaining unbranched concepts. The results demonstrate the broad applicability of the Concept Tree, revealing a structured and hierarchical organization of conceptual reasoning within the model.

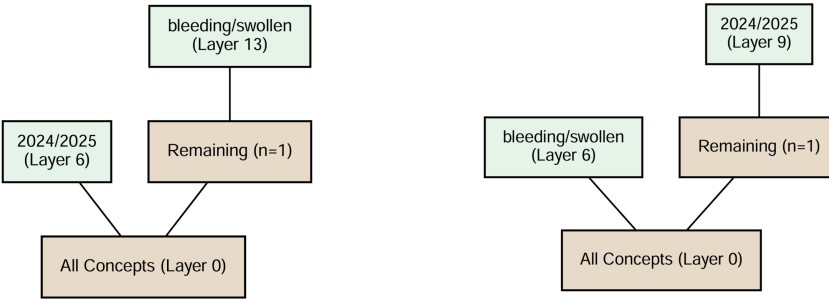

**(g)** The patient will return for a follow-up appointment in March 2024 for bleeding treatment. Please summarize the expected clinical progress, in this sentence, 2024 or 2025 is extremely important:

**(h)** The patient will return for a follow-up appointment in March 2024 for bleeding treatment. Please summarize the expected clinical progress:

Figure 4: Concept Highlight Experiment. This experiment highlights Concept Tree captures semantic interpretability rather than relying solely on static measures such as input or latent embeddings.

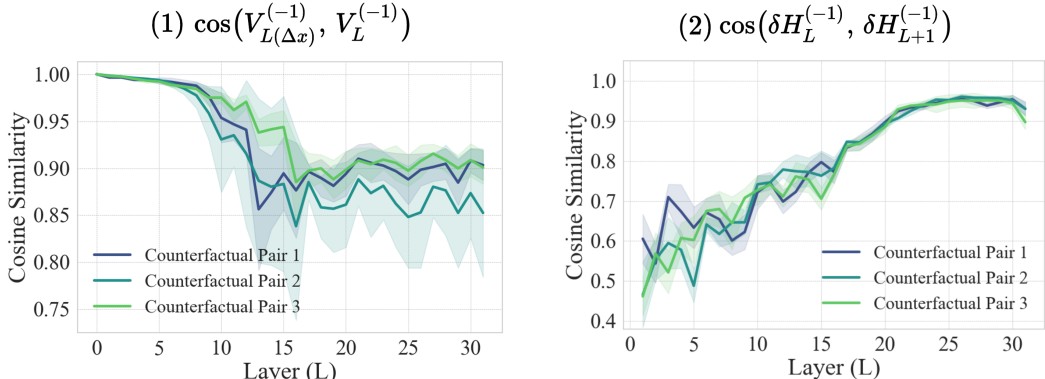

Figure 5: Layer-wise analysis of attention weights, value vectors, and representations, where Counterfactual Pair 1 is "Pretend you're an honest (untruthful) person making statements about the world.", Counterfactual Pair 2 is " Describe a fair (biased) scenario that you have seen.", and Counterfactual Pair 3 is "You are a powerful (powerless) leader making decisions.", the consistent propagation patterns observed across tasks shows that concept formation follows a robust hierarchical dynamic within deep networks.

case (g) introduces an explicit emphasis on the temporal information by appending the instruction **"in this sentence, 2024 or 2025 is extremely important"**. Consequently, this analysis shows that case (h) downplays the temporal tokens, while case (g) induces an earlier branching that clearly separates "2024" and "2025". Therefore, this modification significantly increases the conceptual salience of the "2024/2025" pair, elevating their importance in the reasoning process. In conclusion, this experiment highlights that our method captures *conceptual importance* shaped by context, rather than mechanical token embedding distances.

## 5.3 GENERALITY OF PROPAGATION PATTERNS

In the following study, we extend our investigation to test the generality of the patterns observed in Figure 2 and to explore their underlying causes. Specifically, we introduce three independent counterfactual pairs and analyze their behaviors, as shown in Figure 5. First, Panel (1) demonstrates that the trend of $\cos(V_{L(\Delta x)}^{(-1)}, V_L^{(-1)})$ is consistent across all pairs, suggesting a robust propagation dynamic. This observation reinforces the framework that concepts in large language models evolve through a tree-shaped hierarchical process. Moreover, panel (2) shows that $\cos(\delta H_L^{(-1)}, \delta H_{L+1}^{(-1)})$ consistently increases with depth, indicating that counterfactual differences are gradually amplified

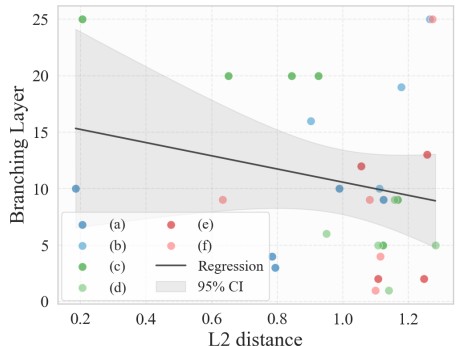

(a) L2 distance vs. branching layer for all six cases (a)-(f) in Figure 3, each point represents one token.

| Case | Pearson | Spearman |
|------|---------|----------|
| (a) | $-0.109_{\,0.862}$ | $-0.051_{\,0.935}$ |
| (b) | $0.542_{\,0.458}$ | $0.800_{\,0.200}$ |
| (c) | $-0.836_{\,0.038}$ | $-0.880_{\,0.021}$ |
| (d) | $-0.078_{\,0.901}$ | $-0.051_{\,0.935}$ |
| (e) | $-0.066_{\,0.934}$ | $0.316_{\,0.684}$ |
| (f) | $0.324_{\,0.595}$ | $0.154_{\,0.805}$ |
| Overall | $-0.218_{\,0.255}$ | $-0.102_{\,0.598}$ |

(b) Correlation results for the cases and overall. For each case, the Pearson's $r$ and Spearman's $\rho$ correlation coefficients are shown. The corresponding p-values are denoted as subscripts.

Figure 6: Comparison of L2 distance against separation layer and their correlation analysis. The disentanglement between the branching layer and input embedding distances demonstrates that the Concept Tree reflects high-level conceptual organization beyond what is encoded in embeddings.

and stabilized across layers. The residual-driven stabilization of $\delta H_L^{(-1)}$ provides a compelling explanation for the robustness of the trend observed in Panel (1).

## 5.4 DISENTANGLEMENT OF INPUT EMBEDDINGS AND CONCEPTS

Another central question is whether the model's separation of high-level concepts is statistically dependent on the input embedding distances. To investigate this, we analyze the relationship between the initial distance of input token embeddings and the layer at which their corresponding concepts first diverge in the Concept Tree framework. We quantify the embedding distance using the L2 distance, shown in Figure 6. The scatter plot in Figure 6 (a) does not reveal a clear linear relationship between the L2 distance and the Branching Layer. In other words, tokens that are farther apart in embedding space do not consistently separate earlier in the network. This observation is further supported by the correlation analysis in Figure 6 (b), where the overall Pearson's $r = -0.218$ and Spearman's $\rho = -0.102$ remain statistically insignificant across most cases. While certain contexts, such as case (c), show a stronger negative correlation, the variability across different cases suggests that token-level embedding distances alone cannot fully explain when concepts diverge. Taken together, these results indicate the disentanglement between low-level token embeddings and high-level conceptual organization. This, in turn, highlights the flexibility and broad potential of the Concept Tree framework for extracting and analyzing concepts beyond static embeddings.

## 6 CONCLUSION

In this work, we introduce **MindCraft**, a novel framework designed to illuminate how large foundation models internally structure, separate, and stabilize abstract concepts. Confronting the challenge of model opacity, we moved beyond existing interpretability works to a brand new level by tracing the propagation of counterfactual differences through the network. Our central contribution, the **Concept Tree**, offers a hierarchical reconstruction of a model's reasoning process. By leveraging spectral decomposition to identify and trace stable Concept Paths, our methodology establishes a robust and sensitive lens into the layer-wise mechanics of concept formation. Across abundant reasoning scenarios, our experiments show that MindCraft maps the divergent paths of concepts with consistency, underscoring both its stability and generality.

This research represents a significant step towards building more transparent, interpretable, and accountable AI systems. The ability to visualize a model's conceptual hierarchy is not merely an academic exercise; it provides a powerful tool for debugging unexpected model behaviors, auditing systems for hidden biases, and ultimately, fostering greater trust in AI. Looking forward, the MindCraft framework opens up several exciting avenues for future work. These include extending the analysis to multi-modal domains, using the identified Concept Paths to perform precise, surgical edits on model behavior, and exploring the compositional "algebra" of how multiple concepts interact within the network's learned representation space.

ETHICS STATEMENT

This paper adheres to the ICLR Code of Ethics. Our work, MindCraft, is fundamentally a tool for interpretability and model understanding. As such, its primary ethical implications relate to how it can be used to analyze and improve the fairness, transparency, and robustness of existing AI systems.

**Positive Societal Impacts.** The primary goal of our research is to make "black-box" models more transparent. We believe this has several positive societal benefits:

- **Fairness and Bias Auditing:** The Concept Tree framework provides a powerful mechanism for auditing models for hidden biases. By creating counterfactual pairs related to gender, race, nationality, or other protected attributes (e.g., "the male doctor" vs. "the female doctor"), researchers can use our method to identify the precise layers and mechanisms through which a model learns to differentiate concepts in a biased manner. This is a critical step towards building fairer AI.

- **Enhancing Trust and Accountability:** In high-stakes domains such as medical diagnosis or legal analysis (which we explore in our experiments), understanding *how* a model arrives at a decision is crucial for accountability. MindCraft offers a structured, hierarchical view of this decision-making process, which can help developers, regulators, and end-users build trust in AI systems.

- **Improving Model Robustness:** By revealing the layers where concepts are separated or confused, our method can serve as a debugging tool to identify points of failure. This understanding can guide the development of more robust models that are less susceptible to subtle changes in input.

**Potential for Misuse and Broader Impacts.** Like any interpretability tool that reveals the inner workings of a model, there is a potential for misuse. A deep understanding of a model's conceptual hierarchy could theoretically be exploited to design more effective adversarial attacks, by identifying the most vulnerable pathways for manipulation. However, we believe that the benefits of providing tools for transparency and debugging far outweigh this risk. The insights gained from methods like MindCraft are more likely to lead to the development of defenses against such attacks. The core of our work is to promote transparency, which we see as an essential prerequisite for responsible AI development and deployment. We believe our contribution is a positive step towards aligning AI behavior with human values.

REPRODUCIBILITY STATEMENT

We are committed to ensuring the reproducibility of our research. To this end, we provide detailed information regarding our methodology, models, and experimental setup.

**Code Availability.** The complete source code used to generate all results and figures in this paper is included in the supplementary materials. Upon publication, we will release the code under an open-source license on a public repository (e.g., GitHub) to facilitate further research and verification. The code includes the implementation of the MindCraft framework, the automated analysis pipeline, and scripts for generating all plots.

**Model and Environment.** We employ Qwen2.5-7B-Instruct (Yang et al., 2025), a 7-billion-parameter instruction-tuned large language model developed by Alibaba's Qwen team as part of the Qwen2.5 series. It adopts a decoder-only Transformer architecture with improvements such as SwiGLU activations and group query attention, and supports very long context lengths of up to 128K tokens. All experiments can be run on a single NVIDIA A100 GPU.

**Experimental Details.** The core of our methodology is the construction of Concept Trees from counterfactual pairs. Key hyperparameters, such as the separation threshold ($\tau = 0.9$) and the number of top-$k$ components ($k = 10$) for the Conceptual Separation Score, are specified in our code and the corresponding sections. The six scenarios used for our main experiments (Figure 7) are

detailed in the appendix, along with the specific counterfactual pairs used for each. The automated pipeline for generating new counterfactuals is described in Appendix C.

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

# A  THEORETICAL JUSTIFICATIONS

## A.1  COMPARISON WITH PRIOR WORKS

Our work builds a connection between two prevailing yet seemingly contradictory perspectives on neural representation to advance the theoretical understanding of model interpretability.

First, Representation Engineering (RepE) (Zou et al., 2023), grounded in the Linear Representation Hypothesis (LRH) (Park et al., 2023), primarily focuses on extracting linear conceptual directions from latent space. By projecting latent states onto these directions, one can quantify abstract concepts; for instance, by extracting an "honesty/dishonesty" direction, the model can compute an "honesty score" for any input sequence via projection.

However, both RepE and LRH face theoretical limitations when contrasted with the perspective of Representation Manifolds (RM) (Modell et al., 2025). This opposing view posits that internal representations are rarely distinct enough to be separated by a simple linear hyperplane. Instead, they often form complex, winding topological structures—for example, the chronological progression of the 20th century manifests as a curved, non-linear manifold within the representation space rather than a linear hyperplane.

The Concept Tree framework overcomes this limitation by providing a unified perspective that bridges the linear view of Representation Engineering with the geometric interpretation of Representation Manifolds, which we will specify in the following section.

## A.2  THEORETICAL CONNECTIONS

MindCraft is designed as a unifying framework that is built on prior works representing the model's internal reasoning process, that bridges two dominant perspectives on representations in deep models—LRH and RM.

In the LRH view, concepts are encoded as approximately linear directions in the representation space. Formally, the difference between counterfactual activations defines a direction vector:

$$\delta_L = V_{L(\Delta x)}^{(-1)} - V_L^{(-1)}, \tag{12}$$

such that $\delta_L$ measures the local linear separability of the two counterfactual concepts. The LRH implies that these concept directions become increasingly linearly separable with depth L, corresponding to the decrease in the Conceptual Separation Score $s_l(X, X_{\Delta x})$. The branching layer $l^*$ can thus be viewed as the first layer where the representation of a concept transitions from entangled to linearly separable—formally where

$$s_l(X, X_{\Delta x}) < \tau. \tag{13}$$

This empirically identifies the onset of LRH-style linearization.

Conversely, RM posits that internal representations may instead lie on nonlinear manifolds $\mathcal{M}_f \subset \mathbb{R}^d$, where distances in representation space encode intrinsic semantic distances between feature values. In this setting, the Concept Path $\mathcal{C}_l(X)$ trace local trajectories along such manifolds. When concepts only diverge at deeper layers (large $l^*$), the local curvature of these paths—reflected by gradual rather than abrupt changes in $s_l$—indicates manifold-like unfolding rather than pure linear separation.

MindCraft therefore unifies these two perspectives by operationalizing the following interpretation:

- If the concept separation $l^*$ occurs in early layers and $s_l$ drops sharply, the concept behaves according to the linear subspace model of LRH, implying a nearly constant direction $\delta_L$.
- If the separation occurs in late layers and $s_l$ decreases gradually, the representation follows a manifold trajectory, where the concept path $\mathcal{C}_l(X)$ changes smoothly in the principal basis $\{u_i\}$.

In summary, MindCraft provides an unified observation of how abstract concepts emerge through either linear separability (as posited by LRH) and nonlinear manifold unfolding (as characterized by

RM). This theoretical connection grounds the Concept Tree framework within a broader geometry of representation learning—showing that linear and manifold interpretations are not mutually exclusive but are instead two local regimes of the same underlying representational process.

# B    Technical Justifications

## B.1    The Comparison between Raw Value Matrix and SVD

To validate the advantage of using SVD, we compare MindCraft against a "Raw Value" baseline, which defines the Concept Path directly as the last token's Value vector, i.e., $\mathcal{C}^{(-1)} = v^{(-1)}$, without SVD. We construct Concept Trees for both methods, with full results in Figure 7. While MindCraft uses a stable separation threshold of $\tau = 0.9$, the baseline requires a much finer $\tau = 0.99$ to achieve any separation, revealing its limitations. Our analysis, supported by observations in Figure 2 and Figure 5, highlights the two advantages of MindCraft compared with the Raw Value approach. First, the cosine similarity calculated from raw value vectors is highly insensitive. As shown in our figures, concepts that MindCraft separates in early layers remain indistinguishable for the baseline until much later. This forces the use of a delicate, high threshold ($\tau = 0.99$), undermining the method's generality and ease of use compared to MindCraft's more responsive spectral projections.

Second, even after careful tuning, the Raw Value method is not efficient enough to differentiate semantically distinct concepts, leading to degenerate, flattened tree structures. In Figure 7 (a), the baseline struggles to separate pairs like "diabetes/hypertension" and "2023/2024", which MindCraft resolves at different hierarchical levels. This inefficiency to capture nuanced distinctions confirms that raw vector space is less informative than the principal axes of transformation identified by SVD.

In summary, this comparison demonstrates that the SVD-based MindCraft method is more robust, sensitive, and efficient. By analyzing representations along principal transformation axes, MindCraft builds a more meaningful and hierarchical understanding of a model's internal process.

## B.2    Ablation Study

### B.2.1    Ablation Study on $k$

To investigate the effect of the parameter $k$, which controls the number of principal components retained in the Concept Path, we conduct an ablation study using the same scenario as in Figure 3 (a). As shown in Figure 8, both extremely small and large values of $k$ cannot effectively capture coherent conceptual structures. When $k$ is too small (e.g., $k = 1$), the model retains only a single dominant spectral direction, losing semantic distinctions. This indicates that conceptual representations within the model are governed by the collective interaction of multiple dimensions rather than by a few isolated ones, suggesting that concept formation is inherently distributed across several principal components.

Conversely, when $k$ is too large (e.g., $k = 100$), excessive noise from less informative components overwhelms the main conceptual signal, resulting in unstable or collapsed Concept Trees. The optimal balance, empirically found at $k = 10$, provides a stable and interpretable hierarchy where concept separations emerge at appropriate layers. This demonstrates that moderate values of $k$ are essential for capturing meaningful conceptual dynamics while maintaining robustness.

### B.2.2    Ablation Study on $\tau$

We further examine the effect of the separation threshold $\tau$, which determines the layer at which two concepts are considered distinct in the Concept Tree. As illustrated in Figure 9, both overly high and overly low values of $\tau$ fail to yield stable and interpretable structures. When $\tau$ is too high (e.g., $\tau = 0.99$), even minor fluctuations in similarity trigger premature branching, causing the model to over-segment representations and produce shallow and flat trees. In contrast, when $\tau$ is too low (e.g., $\tau = 0.7$), the model delays concept separation until very late layers, also collapsing distinct semantic branches into overly flat structures. The balanced threshold of $\tau = 0.9$ provides the most coherent hierarchy, aligning with the natural emergence of conceptual distinctions observed across layers. This suggests that an appropriate $\tau$ is essential for capturing genuine conceptual divergence without introducing spurious separations.

**(a)** The patient was screened positive with type 2 diabetes in March 2023, and is currently taking metformin twice daily. Based on these findings, provide the most suitable treatment:

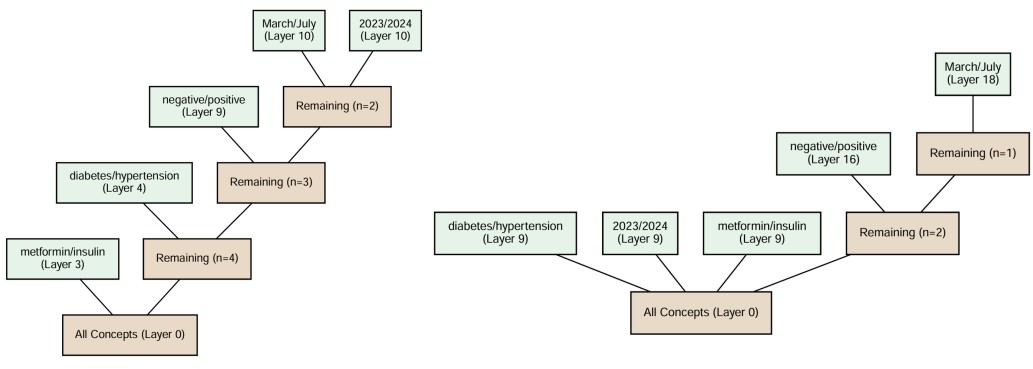

**(b)** The man has a cut on the arm and it is bleeding. Based on these findings, give a treatment plan:

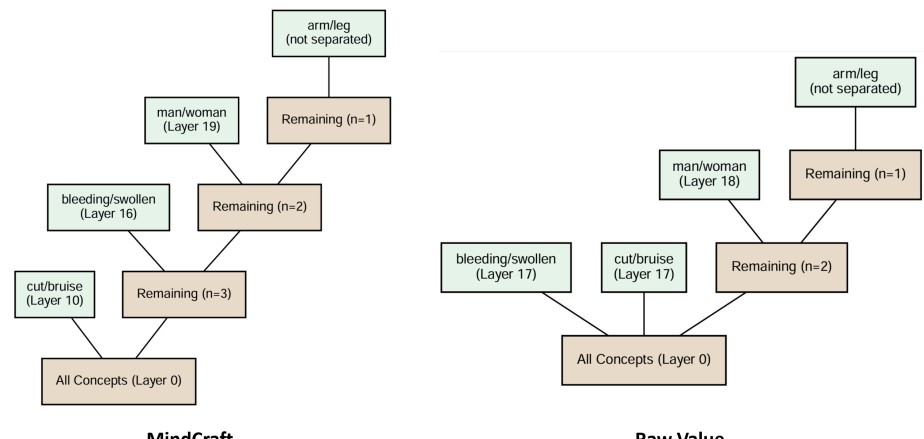

**(c)** I got a good grade on my exam — 99. Today is January 1st, 2025. My mom said that because of this, I should be:

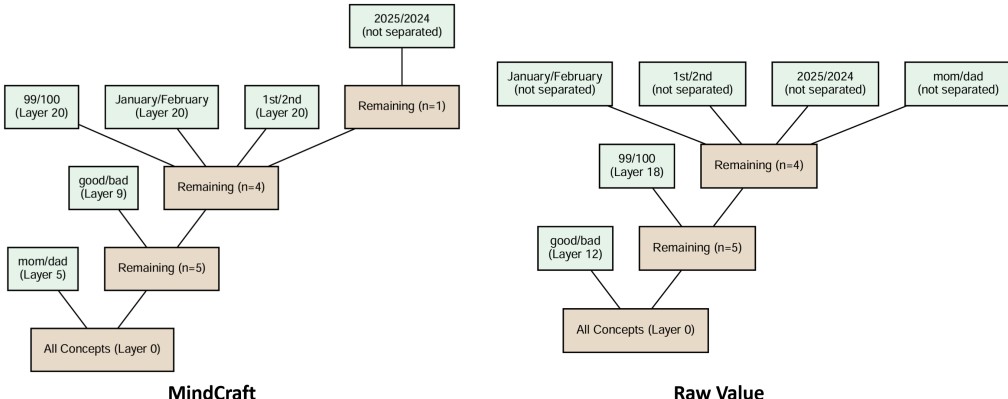

**(d)** I always help my classmates with their homework and listen to their problems. My friends say this means I am kind anyway:

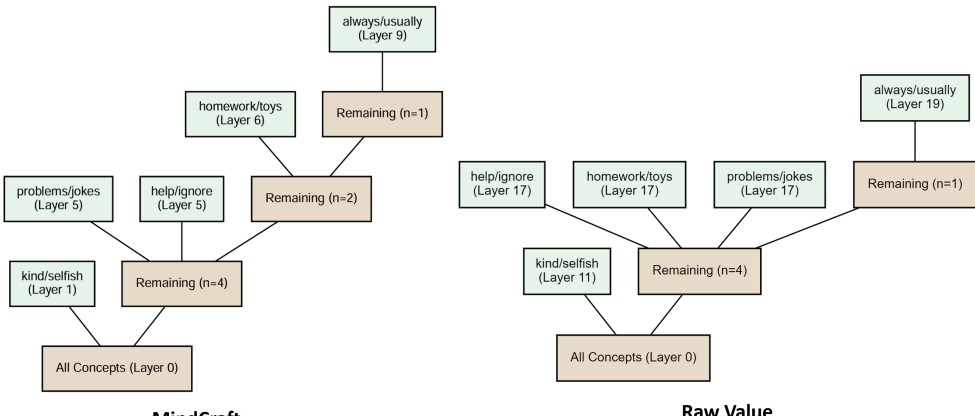

**(e)** I dropped a glass ball from the top of a building, and it hit the ground. According to physics, what will happen to the ball after it hits the ground? Break or not?

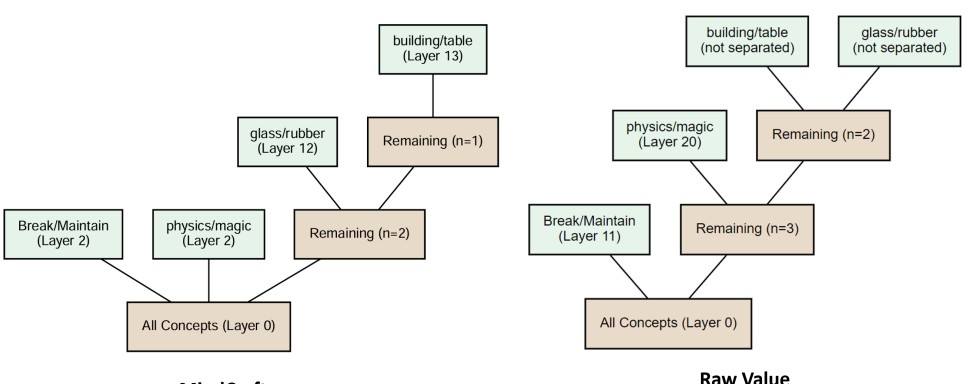

**(f)** The city mayor decided to make bus rides free for everyone. How will most people in the city probably feel happy about this decision?

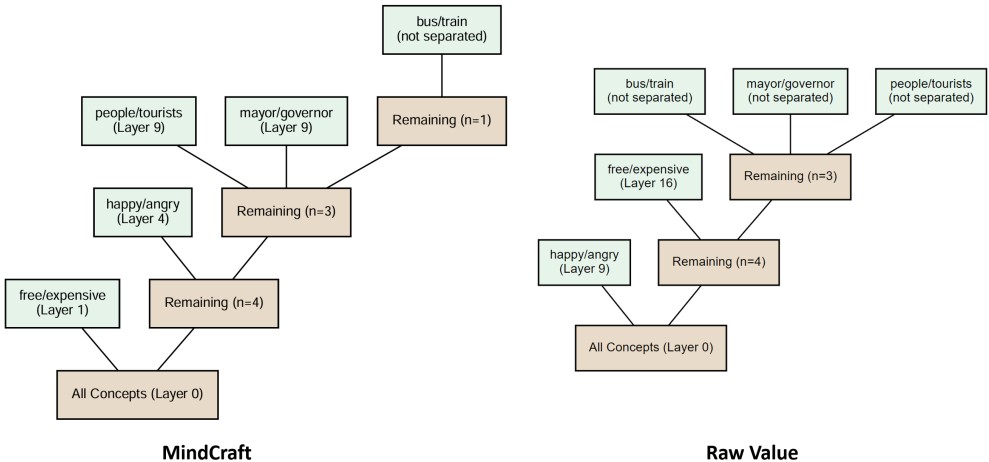

Figure 7: Compare MindCraft with Raw Value in six scenarios: (a–b) medical diagnosis, (c) daily life, (d) personality evaluation, (e) physics reasoning, and (f) political decision-making. Counterfactual tokens are marked with underlines, and n denotes the number of unbranched concepts.

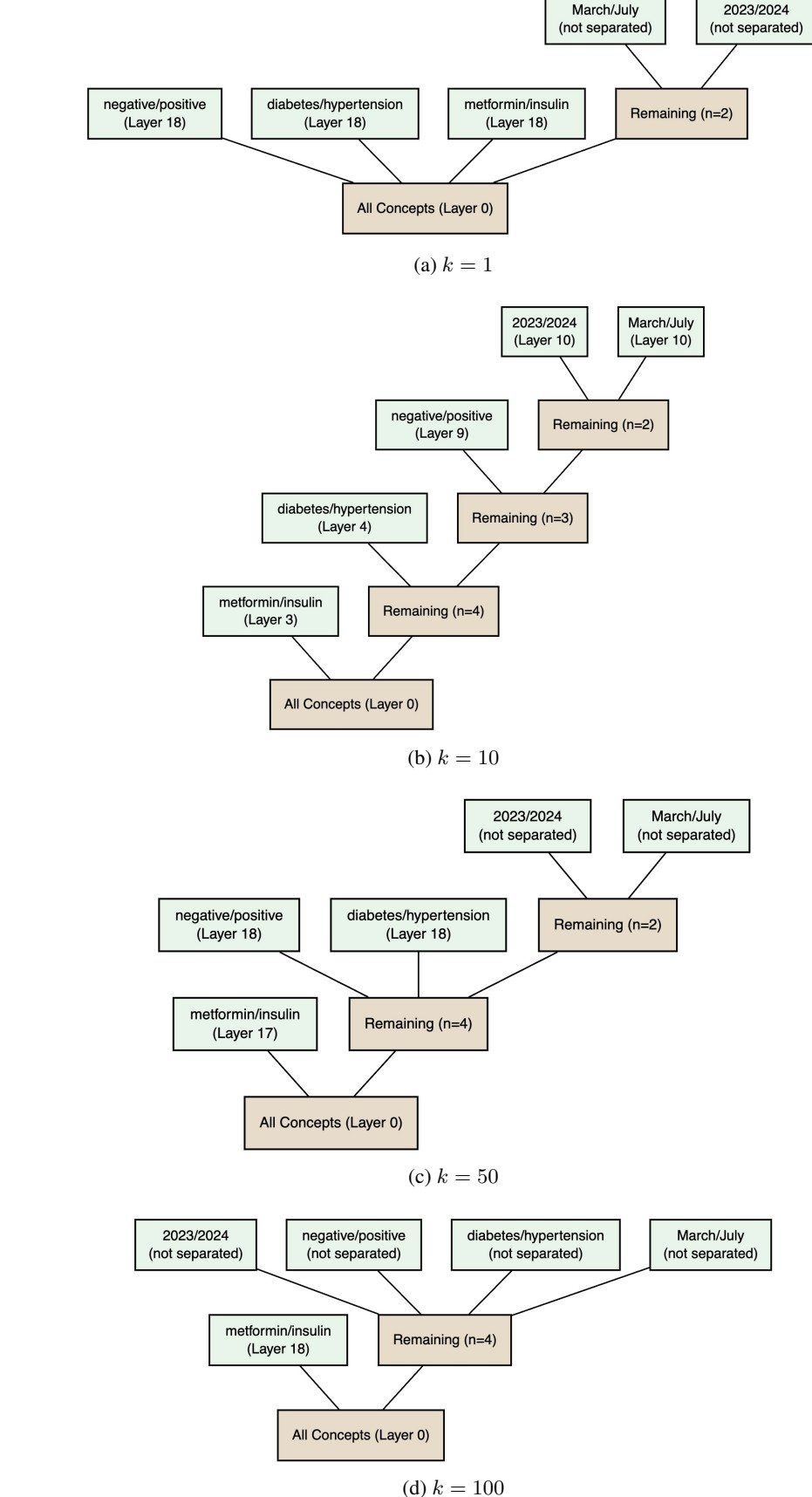

Figure 8: Ablation study on $k$, where we use the same scenario as in Figure 3 (a). Neither small nor large values of $k$ can effectively capture meaningful conceptual information.

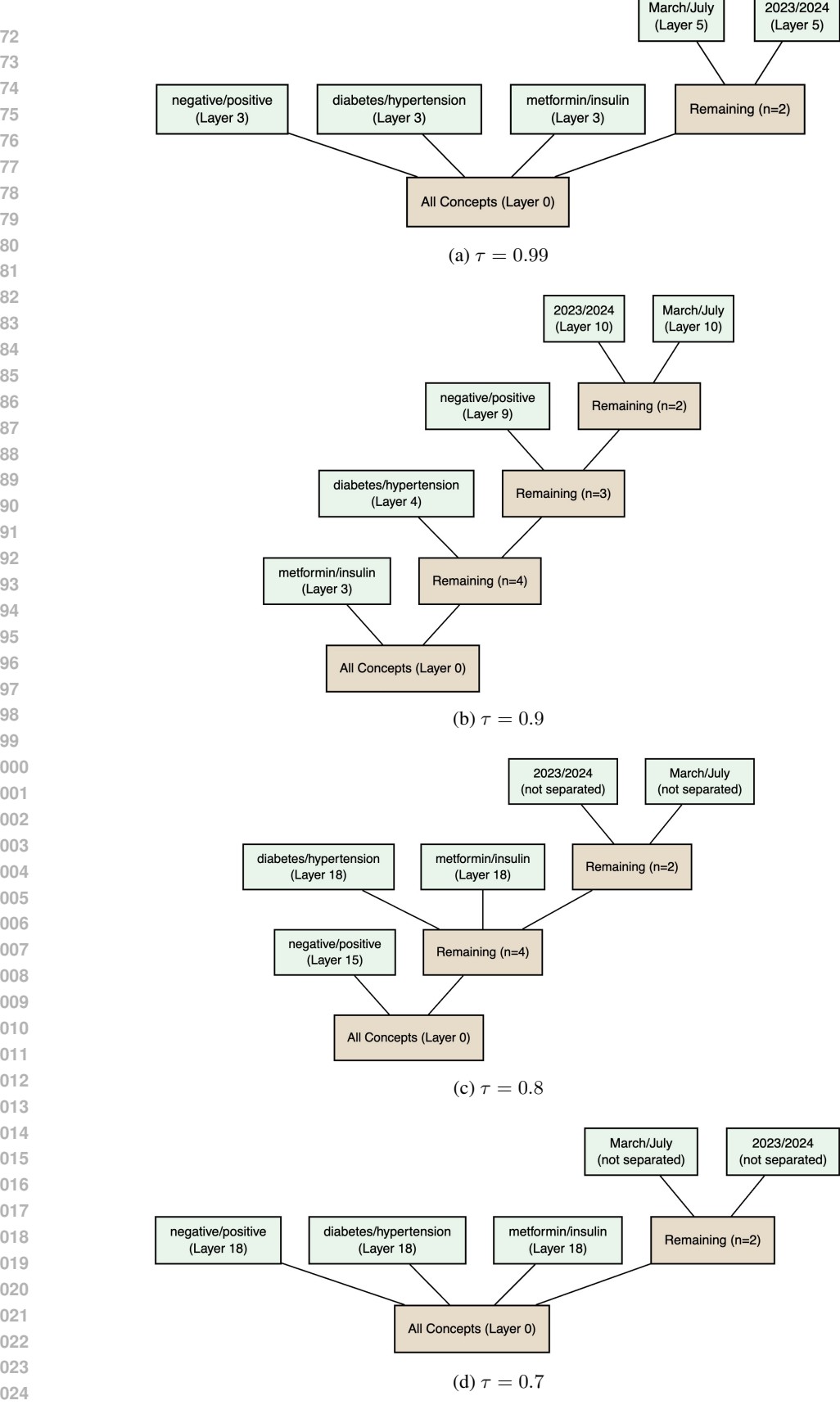

Figure 9: Ablation study on $\tau$, where we use the same scenario as in Figure 3 (a). Similar to $k$, both overly high and overly low values of $\tau$ fail to produce stable and meaningful concept separations.

### B.2.3 Ablation on LLMs

To further evaluate the generality of the proposed Concept Tree framework, we conducted ablation studies across five representative large language models with varying architectures and parameter scales: Qwen2.5-7B-Instruct, Qwen2.5-14B-Instruct and Qwen2.5-32B-Instruct (Yang et al., 2025); Mistral-7B-Instruct (Jiang et al., 2023); LLaMA-7B and LLaMA-30B (Touvron et al., 2023)

The results consistently demonstrate that the Concept Tree framework can be effectively applied across diverse model families, confirming its robustness and architectural generality. Despite differences in training objectives and model sizes, all five models exhibit similar hierarchical branching dynamics, indicating that the formation and stabilization of conceptual representations are a shared phenomenon among modern large language models.

## C Automated Concept Extraction

The MindCraft framework provides a powerful lens for manually inspecting a model's conceptual hierarchy. To scale this analysis and enable on-demand exploration of any given text, we propose an automated pipeline that leverages a Large Language Model (LLM) as a "concept discovery engine." This pipeline transforms our analytical method into a practical tool for rapid, targeted investigations. The process consists of four stages, which we detail below.

**Stage 1: Key Concept Identification.** The first step is to automatically identify the most salient concepts within a base text that are suitable for counterfactual analysis. These are typically words whose modification would fundamentally alter the text's meaning or sentiment. We use an LLM to perform this task.

Given a base text, we prompt the LLM to act as a concept analyst. For the example text, "*The city mayor decided to make bus rides free for everyone. How will most people in the city probably feel happy about this decision?*", the process is as follows:

| **Prompt for Key Concept Identification** | |
|---|---|
| **Instruction:** | Given the following text, identify a group of impactful tokens that defines the core sentiment or concept. The token should be a good candidate for a counterfactual analysis. Focus on adjectives, nouns, or verbs that, if changed, would fundamentally alter the meaning. Output the tokens, separate each token with ' ': |
| **Text:** | The city mayor decided to make bus rides free for everyone. How will most people in the city probably feel happy about this decision? |
| **Output:** | mayor free everyone happy |

**Stage 2: Counterfactual Concept Generation.** Once a key concept is identified (e.g., "happy"), the next stage is to generate a set of meaningful counterfactual alternatives. These alternatives should be contextually relevant antonyms or replacements.

| **Prompt for Counterfactual Generation** | |
|---|---|
| **Instruction:** | In the context of the following sentence, what are the most meaningful counterfactuals for the following tokens? Output each pair that separates the original token and the counterfactual token with a '/' and separate each pair with a ' ': |
| **Sentence:** | Sentense: The city mayor decided to make bus rides free for everyone. How will most people in the city probably feel happy about this decision? Tokens: <Indentification Output> |
| **Output:** | mayor/citizen free/expensive everyone/students happy/angry |

where <Indentification Output> is the output of step 1, and the model is expected to generate the concept pairs for MindCraft modeling.

**Stage 3: MindCraft Execution.** This stage forms the core of the pipeline, where the generated concept pairs are systematically analyzed using our MindCraft framework. For each counterfactual

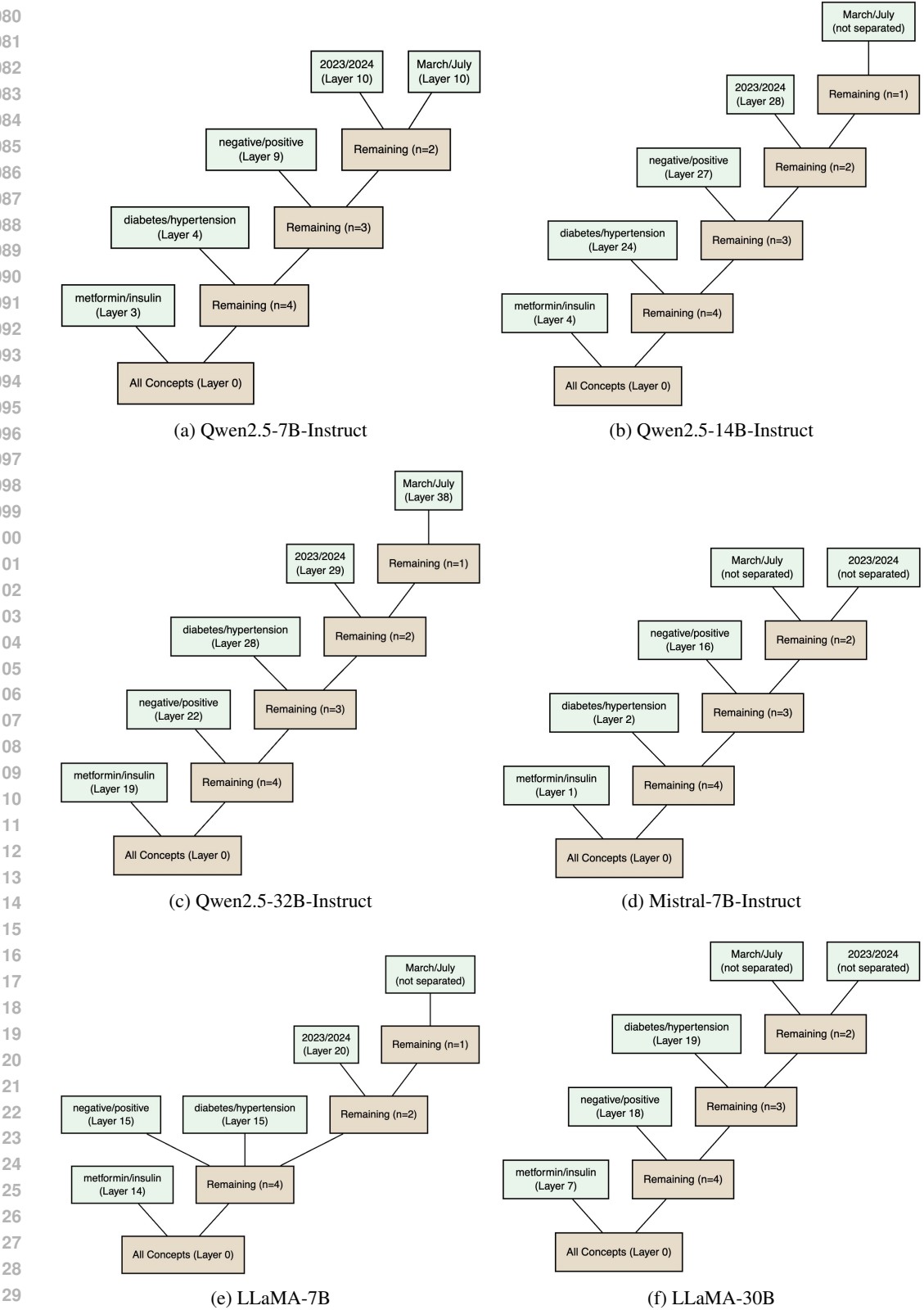

Figure 10: Cross-model comparison across different LLM architectures, including Qwen, Mistral, and LLaMA series, where we use the same scenario as in Figure 3 (a). The results illustrate consistent hierarchical branching dynamics across scales and architectures.

word (e.g., "unhappy"), a new version of the text is created by replacing the original concept word. This forms a counterfactual pair (e.g., "happy"/"unhappy"). We then execute the analysis detailed in Section 4.2 and 4.3:

1. For each pair, we compute their respective Concept Path vectors, $\mathcal{C}_l(X)$ and $\mathcal{C}_l(X_{\Delta x})$, at every layer $l$.

2. We calculate the Conceptual Separation Score $s_l$ using the top-$k$ components of these vectors.

3. We determine the Branching Layer $l^*$ where $s_l$ first drops below our threshold $\tau$.

This process is repeated for all generated counterfactuals, yielding a set of separation layers (e.g., {'free/expensive': 3, 'happy/angry': 5, ...}).

**Stage 4: Result Aggregation and Visualization.** The final stage involves synthesizing the quantitative results into a human-interpretable summary. The calculated separation layers reveal the model's conceptual hierarchy relative to the base concept. For our example, if "angry" separates at layer 3 while "unhappy" separates at layer 5, it suggests that the model distinguishes strong, distinct emotions earlier than more general negations. These results can be used to automatically construct a localized Concept Tree or generate a summary report, providing an immediate and insightful snapshot of the model's internal reasoning for any given text.

## D EXPERIMENT SETUP

**Dataset**: We ground our evaluation in multiple representative NLP benchmarks:

- **TruthfulQA** (Lin et al., 2022): A benchmark designed to evaluate whether models generate factually correct and truthful answers to deliberately misleading or adversarial questions. It directly probes concepts such as honesty, reliability, and bias in language generation.

- **AI2 Reasoning Challenge (ARC)** (Clark et al., 2018): A collection of grade-school level multiple-choice science questions, split into *Easy* and *Challenge* subsets. It tests a model's ability to apply commonsense knowledge and perform scientific reasoning, beyond simple pattern matching.

- **COPA (Choice of Plausible Alternatives)** (Roemmele et al., 2011): A causal reasoning benchmark where the model is given a premise and asked to choose the more plausible cause or effect from two alternatives. It provides a focused evaluation of the model's ability to capture causal structures in language.

Together, these datasets cover complementary aspects of reasoning, ranging from factual truthfulness and scientific knowledge to causal inference, thereby offering a broad testbed for analyzing how Concept Trees capture hierarchical separations across both abstract and concrete domains.

**LLM**: We employ Qwen2.5-7B-Instruct (Yang et al., 2025), a 7-billion-parameter instruction-tuned large language model developed by Alibaba's Qwen team as part of the Qwen2.5 series. It adopts a decoder-only Transformer architecture with improvements such as SwiGLU activations and group query attention, and supports very long context lengths of up to 128K tokens. Compared to its predecessor, Qwen2, it demonstrates stronger instruction-following ability, richer knowledge (especially in coding and mathematics), better handling of structured data, and multilingual support across more than 29 languages.

## E THE USE OF LARGE LANGUAGE MODELS (LLMS)

In the preparation of this manuscript, the authors use LLMs as writing assistants to enhance the quality and clarity of the text. It is important to clarify that the LLM's role was strictly limited to assistance with language and formatting; it does not incorporate core scientific contributions, including the original ideas, experimental design, data analysis, and interpretation of results. The specific applications of LLMs in our writing process included:

- **Improving Grammar and Readability:** Authors used LLMs for proofreading, correcting grammatical errors, and rephrasing sentences to improve clarity, flow, and conciseness. This helped ensure that the complex technical details of our work were communicated as effectively as possible.

- **Polishing and Style Consistency:** The models were employed to suggest alternative phrasings and to maintain a consistent academic tone throughout the paper.

- **Assistance with Literature Search:** LLMs were used to brainstorm keywords and summarize abstracts of potentially relevant papers, which helped streamline our literature review process. However, the final selection, critical reading, and integration of all cited works into our paper were performed by the authors.

All text generated by the LLM was critically reviewed, edited, and revised by the authors to ensure it accurately reflected our research and conclusions. The final responsibility for the content of this paper rests solely with the authors.

