# OpenReview forum: "MindCraft: How Concept Trees Take Shape In Deep Models"
_ICLR.cc/2026/Conference — Submitted to ICLR 2026_

### Official Review · Reviewer_cnAx · 2025-10-29

**Soundness:** 2
**Presentation:** 3
**Contribution:** 2
**Rating:** 6
**Confidence:** 3

**Summary:**

The authors investigate how various concepts "appear" within machine learning models through the lens of causal inference. In particular, they introduce a framework, called MindCraft, which investigates how a pair of counterfactual statements (e.g., "You are a powerful leader" and "You are a powerless leader") differ in last-token representation as we change the layer number. This essentially serves as a signature for a given concept. This can then be used to identify when concepts diverge by placing a threshold for when two concepts should differ. This leads to the creation of trees which visualize branching in this system. The paper demonstrates this for various applications, along with an exploratory experiment that explores whether initial embedding distance is predictive of the separation between high-level concepts.

**Strengths:**

1. **Visually Appealing Representation** - The authors present a tree-based diagram visualizing how different concepts emerge in deep learning models. The representation is visually appealing, and makes it clear the order in which these concepts occur. Such a representation could be valuable, as it allows us to better understand the dynamics underlying these deep models.
2. **Extensive Examples** - The authors present the material in a fairly clear way through the extensive use of examples. For example, page 7 presents numerous examples of concepts emerging in this context, which makes it clear how such trees operate. Moreover, the paper is generally well-presented, with various examples of how concepts emerge, and the meaning of such emergence.
3. **Interesting Exploratory Experiments** - In Section 5, the authors explore the disentanglement of input embeddings from concepts, essentially looking at whether token-level embeddings can predict which concepts emerge. Such an experiment is interesting because it investigates the relationship between model representation with the data itself, to identify whether such concept emergence is inevitable.

**Weaknesses:**

1. **Unclear Generalization** - While the authors present numerous examples throughout the paper, my biggest worry is that the patterns seen here might not generalize. Specifically, my worry is two-fold a) concepts might not always be suddenly amplified, yielding some of their analysis moot, and b) it is unclear how to interpret or use the presented concept trees in practice. What is the insight useful for/how can we interpret this insight?
2. **Lack of Justification for Why** - The authors provide little theoretical justification for why certain layers might propagate signals for concepts moreso than other layers. Many of the plots in the experiments section are for individual examples (which helps to get the point across), but the lack of aggregate analysis makes it hard to understand whether such trends hold across settings, and moreover, why such trends should even hold in the first place.

**Questions:**

1. How should MindCraft be used in practice?

---

> ### Author Response · Authors · 2025-11-23
>
> Thank you for your feedback. Please let us address W2 first:
>
> ### W2. Theoretical justification for why certain layers might propagate signals for concepts more so than other layers
>
> To explain why certain layers might propagate signals for concepts more so than other layers, we introduce the prior works: the Linear Representation Hypothesis (LRH) [1] and the Representation Manifold (RM) [2], and show our theoretical justifications.
>
> Specifically, MindCraft is a framework that *is built on prior works supporting this "branch-and-stabilize" trajectory*, representing the model's internal reasoning process, and bridges two dominant perspectives on representations in deep models—the **Linear Representation Hypothesis (LRH)** [1] and **the Representation Manifold (RM)** [2].
>
> In the **LRH** view, concepts are encoded as approximately linear directions in the representation space. Formally, the difference between counterfactual activations defines a direction vector:
> $$
> \delta_L = V^{(-1)}_{L(\Delta x)} - V^{(-1)}_L,
> $$
>
> such that $\delta_L$ measures the local linear separability of the two counterfactual concepts. The LRH implies that these concept directions become increasingly linearly separable with depth L, corresponding to the decrease in our Conceptual Separation Score $s_l(X, X_{\Delta x})$ (Eq. 10). The *branching layer* $ l^*$ (Eq. 11) can thus be viewed as the first layer where the representation of a concept transitions from entangled to linearly separable—formally where
> $$
> s_{l}(X, X_{\Delta x}) < \tau.
> $$
>
> This empirically identifies the onset of LRH-style linearization.
>
> Conversely, **RM** posits that internal representations may instead lie on *nonlinear manifolds* $\mathcal{M}_f \subset \mathbb{R}^d$, where distances in representation space encode intrinsic semantic distances between feature values.
> In this setting, the Concept Path $\mathcal{C}_l(X)$ (Eq. 6) trace local trajectories along such manifolds. When concepts only diverge at deeper layers (large $ l^* $), the local curvature of these paths—reflected by gradual rather than abrupt changes in $s_l$—indicates manifold-like unfolding rather than pure linear separation.
>
> MindCraft therefore **unifies these two perspectives** by operationalizing the following interpretation:
>
> - If the concept separation $l^*$ occurs in *early* layers and $s_l$ drops sharply, the concept behaves according to the **linear subspace model** of LRH, implying a nearly constant direction $\delta_L$.
> - If the separation occurs in *late* layers and $s_l$ decreases gradually, the representation follows a **manifold trajectory**, where the concept path $\mathcal{C}_l(X)$ changes smoothly in the principal basis $\\{u_i\\}$.
>
> In summary, **MindCraft** provides an unified observation of how abstract concepts emerge through either *linear separability* (as posited by LRH) and *nonlinear manifold unfolding* (as characterized by RM). This theoretical connection explains why certain layers might propagate signals for concepts more so than other layers. We provide more details in Appendix A.
>
> ### W1. a) concepts might not always be suddenly amplified, yielding some of their analysis moot
>
> Based on the reasoning of W2, we can provide a more comprehensive answer to your question.
>
> Empirically, we observe that the vast majority of concepts indeed follow this "branch-and-stabilize" pattern. The primary variation lies in the magnitude of the decline in similarity (as illustrated by the red bars in Figure 2 and the sharp drop in Figure 5 (1) ).
>
> Theoretically, our analysis builds upon prior work regarding the geometry of representation space—specifically the debate between the Linear Representation Hypothesis (LRH) and the Representation Manifold (RM) perspective. Our framework unifies these views (as detailed in Appendix A.2):
>
> A sharp, sudden drop in similarity indicates that the concept forms a linearly separable subspace (consistent with LRH).
>
> A more gradual decline suggests the concept follows a nonlinear manifold trajectory (consistent with RM).
>
> Both scenarios, however, fall within the broader "branching" behavior defined by our framework.
>
> A true deviation from this dynamic would be a scenario exhibiting extreme volatility—where similarity drops to near 0 and immediately bounces back to near 1. Such behavior would indeed challenge the current works of internal representations. However, to date, we have not observed such high-volatility concepts in our experiments. Consequently, we conclude that the "branch-and-stabilize" pattern is both empirically robust and theoretically stable.

---

> ### Author Response · Authors · 2025-11-23
>
> ### W1. b) & Q1:  it is unclear how to interpret or use the presented concept trees in practice. What is the insight useful for/how can we interpret this insight?
>
> We expect the Concept Tree to be a widely used tool that begins with visualizations of neural networks' black box and develops into an intelligent controller (for example, we begin from simple concepts such as "honesty/dishonesty", but we will continue to explore deeper concepts, such as "faithful/skeptic" that can (we believe that will be) deep enough to recognize a machine's personality and belief, and eventually self-consciousness). Having said that, we can see several applications at this stage:
>
> ### Use Case A: Verifying Model Reasoning (The "Right for the Right Reasons" Test)
> We often know that a model got the answer right, but not why. The Concept Tree reveals the priority list the model used to reach the conclusion.
>
> Insight: In the Medical Diagnosis (Figure 3a), a doctor might worry: "Is the model prescribing Metformin because it sees 'Diabetes', or because it sees the date '2023'?"
>
> The Tree's Answer: The Tree shows metformin/insulin branches at Layer 3, while 2023/2024 branches at Layer 10.
>
> Conclusion: The model correctly prioritized the medical condition over the date. If the date had branched at Layer 3, you would immediately know the model was over-relying on irrelevant temporal features (a "shortcut" bug).
>
> ### Use Case B: Prompt Engineering & Control Verification
> You can use the tree to verify if your prompt actually changed the model's internal behavior.
>
> Insight: In the Concept Highlight Experiment (Figure 4), you essentially ask: "If I tell the model that the date is important, does it actually listen?"
>
> The Tree's Answer:
>
> Without instruction: 2024/2025 branches late (Layer 9).
>
> With instruction ("Date is extremely important"): 2024/2025 branches early (Layer 6).
>
> Conclusion: The Tree provides quantitative proof that the prompt successfully "promoted" the concept of time to a higher priority level in the model's internal hierarchy. This allows developers to A/B test prompts scientifically.
>
> ### Use Case C: Fairness and Bias Auditing
> The Concept Tree serves as a bias detector by revealing "silent" distinctions.
>
> Insight: In a hiring scenario, does the model distinguish between "Male Candidate" and "Female Candidate"?
>
> The Tree's Answer:
>
> Fair Model: The pair male/female should branch very late or be not separated (meaning the model treats them identically for the task).
>
> Biased Model: If male/female branches at Layer 2, the model is treating gender as a fundamental, deciding factor—even if the final text output looks neutral.
>
> Conclusion: The branching layer serves as a "Bias Severity Score." The earlier the branch, the deeper the bias.
>
> [1] Park et al. The Linear Representation Hypothesis and the Geometry of Large Language Models. arXiv preprint arXiv:2311.03658, 2024.
>
> [2] Modell et al. The Origins of Representation Manifolds in Large Language Models. arXiv preprint arXiv:2505.18235, 2025.

---

### Official Review · Reviewer_ZdDy · 2025-10-30

**Soundness:** 1
**Presentation:** 2
**Contribution:** 2
**Rating:** 2
**Confidence:** 3

**Summary:**

To address opacity of neural networks, the paper introduces MindCraft, a method to inspect how the internal representation of an abstract concept evolves through the layers of a neural network. To do so, it creates counterfactual pairs and tracks internal representation differences between at various depths of a foundation model, identifying where concepts diverge. MindCraft can also provide interpretable visualizations for model development and debugging. The reported experiments and visualizations suggest generalization across domains.

**Strengths:**

- Novelty: I like the idea of studying how the internal representation of an abstract concept evolves across different layers. A tool capable of performing such analysis could be valuable, as it enables a more comprehensive understanding of the model itself.

**Weaknesses:**

Major:

- **W1** - The authors claim that MindCraft explains how large foundation models internally structure abstract concepts (l. 470–472). However, according to the experimental section, the proposed methodology is tested on only one LLM (Qwen2.5-7B-Instruct). Therefore, the statement in the conclusion lacks sufficient empirical support. Without this claim, the overall impact of the paper is considerably reduced.

- **W2** - The results presented in the experimental section are mostly qualitative (Figures 3-4-5). Although the authors provide several qualitative examples, a formal validation (established metrics and/or statistical tests) supporting the consistency and reliability of the approach appears to be missing. This limitation likely stems from the absence of a theoretical ground behind both the motivation and the methodology.


Minor:

- **W3** - Motivations seem weak. In l.194, the authors claim “Concept formation, therefore, follows a branch-and-stabilize process:” and, later in l.198 “concept-level organization is not static, but unfolds progressively through the network.” At this point of the paper, this is supported only using a single example (Fig.2). Also, I would expect the validity and extent of such a general claim, as well as the particular observed dynamics of the similarity score, to depend strongly on the specific examples, i.e., sentences and context.

- **W4** - The necessity of using the principal directions extracted from the SVD with respect to the raw ( $W_V$ ) is not well justified. Despite showing several examples in the appendix, the rationale for this choice remains primarily qualitative.

- **W5** - The concept tree is constructed considering only self-attention. However, a generic LLM is far more complex than that, as LLMs typically employ multi-head attention. Consequently, the internal representation of a specific sequence at a given depth is only partially analyzed by the proposed methodology. Shouldn't it also account for the other attention heads at the same depth?

**Questions:**

- **Q1** (related to W3) - Have you observed any scenario where the dynamics of the counterfactual pair is not “branch-and-stabilize“?

- **Q2** (related to W5): How do you think your methodology extends to or interacts with multi-head attention or more complex architectures?

While I remain open to a constructive discussion, I believe the paper requires substantial improvement, especially towards establishing a clearer/stronger theoretical foundation for both the proposed method and its evaluation. At this stage, the gap between the current submission and a version that would meet the bar for acceptance still feels wide to me.

---

> ### Author Response · Authors · 2025-11-23
>
> ### W1: Empirical Support and Generalization across LLMs
>
> We agree that validating our findings across diverse architectures is crucial for the generalizability. To address this, we have expanded our experiments in Appendix B.2.3 . In Figure 10, we have included a cross-model comparison across representative LLMs with varying architectures and parameter scales:
>
> - Qwen Series: Qwen2.5-7B-Instruct, Qwen2.5-14B-Instruct, and Qwen2.5-32B-Instruct.
>
> - Mistral Series: Mistral-7B.
>
> - LLaMA Series: LLaMA-7B and LLaMA-30B.
>
> Using the same experimental scenario as in Figure 3(a), the results consistently illustrate that hierarchical dynamics are preserved across different model families and scales, providing empirical support that the Concept Tree captures a fundamental mechanism of concept formation rather than an artifact of a specific architecture.
>
> ### W2. On the Formal Metric and Quantitative Validation
>
> While our visualizations emphasize interpretability, MindCraft introduces a quantitative metric — the Conceptual Separation Score (Eq. 10) — which measures the cosine similarity between concept-path projections across layers. This score enables the definition of the Branching Layer $l^*$, where conceptual divergence first emerges, forming the quantitative foundation of the Concept Tree.
> We further validate this metric’s robustness through correlation analysis (Figure 6) and ablation studies (Appendix B), showing its stability across tasks, hyperparameters, and models, ensuring a balanced and rigorous assessment.
>
> ### W3: Motivation and Generality of "Branch-and-Stabilize"
>
> Thank you for pointing this out. We clarify that the example in Section 4.1 was intended as an illustration to help readers quickly grasp the intuition of the Concept Tree. To demonstrate that this is a general phenomenon rather than a context-specific anomaly, we provide the following evidence:
>
> - Empirical Generalization: As shown in Figure 5 (1), we analyze the cosine similarity trends across three independent counterfactual pairs and contexts. The results show a consistent "branch-and-stabilize" trajectory across all cases—similarity remains high in early layers, drops sharply at the branching point, and stabilizes in deeper layers. This confirms that the dynamics are robust to different sentences and semantic contexts.
>
> - Theoretical Grounding: MindCraft is designed as a unifying framework that is built on prior works representing the model's internal reasoning process, that bridges two dominant perspectives on representations in deep models—the **Linear Representation Hypothesis (LRH)** [1] and **the Representation Manifold (RM)** [2].
>
> In the **LRH** view, concepts are encoded as approximately linear directions in the representation space. Formally, the difference between counterfactual activations defines a direction vector:
> $$
> \delta_L = V^{(-1)}_{L(\Delta x)} - V^{(-1)}_L,
> $$
>
> such that $\delta_L$ measures the local linear separability of the two counterfactual concepts. The LRH implies that these concept directions become increasingly linearly separable with depth L, corresponding to the decrease in our Conceptual Separation Score $s_l(X, X_{\Delta x})$ (Eq. 10). The *branching layer* $ l^*$ (Eq. 11) can thus be viewed as the first layer where the representation of a concept transitions from entangled to linearly separable—formally where
> $$
> s_{l}(X, X_{\Delta x}) < \tau.
> $$
>
> This empirically identifies the onset of LRH-style linearization.
>
> Conversely, **RM** posits that internal representations may instead lie on *nonlinear manifolds* $\mathcal{M}_f \subset \mathbb{R}^d$, where distances in representation space encode intrinsic semantic distances between feature values.
> In this setting, the Concept Path $\mathcal{C}_l(X)$ (Eq. 6) trace local trajectories along such manifolds. When concepts only diverge at deeper layers (large $ l^* $), the local curvature of these paths—reflected by gradual rather than abrupt changes in $s_l$—indicates manifold-like unfolding rather than pure linear separation.
>
> MindCraft therefore **unifies these two perspectives** by operationalizing the following interpretation:
>
> - If the concept separation $l^*$ occurs in *early* layers and $s_l$ drops sharply, the concept behaves according to the **linear subspace model** of LRH, implying a nearly constant direction $\delta_L$.
> - If the separation occurs in *late* layers and $s_l$ decreases gradually, the representation follows a **manifold trajectory**, where the concept path $\mathcal{C}_l(X)$ changes smoothly in the principal basis $\\{u_i\\}$.
>
> In summary, **MindCraft** provides an unified observation of how abstract concepts emerge through either *linear separability* (as posited by LRH) and *nonlinear manifold unfolding* (as characterized by RM). This theoretical connection grounds the Concept Tree framework within a broader geometry of representation learning. We provide details in Appendix A.

---

> ### Author Response · Authors · 2025-11-23
>
> ### Q1: Have you observed any scenario where the dynamics of the counterfactual pair is not “branch-and-stabilize“?
>
> Based on the reasoning of W3, we can provide a more comprehensive answer to your question.
>
> Empirically, we observe that the vast majority of concepts indeed follow this "branch-and-stabilize" pattern. The primary variation lies in the magnitude of the decline in similarity (as illustrated by the red bars in Figure 2 and the sharp drop in Figure 5 (1)).
>
> Theoretically, our analysis builds upon prior work regarding the geometry of representation space—specifically the debate between the Linear Representation Hypothesis (LRH) and the Representation Manifold (RM) perspective. Our framework unifies these views (as detailed in Appendix A.2):
>
> A sharp, sudden drop in similarity indicates that the concept forms a linearly separable subspace (consistent with LRH).
>
> A more gradual decline suggests the concept follows a nonlinear manifold trajectory (consistent with RM).
>
> Both scenarios, however, fall within the broader "branching" behavior defined by our framework.
>
> A true deviation from this dynamic would be a scenario exhibiting extreme volatility—where similarity drops to near 0 and immediately bounces back to near 1. Such behavior would indeed challenge the current works of internal representations. However, to date, we have not observed such high-volatility concepts in our experiments. Consequently, we conclude that the "branch-and-stabilize" pattern is both empirically robust and theoretically stable.
>
> ### W4: Justification for SVD vs. Raw Value ($W_V$)
>
> We have strengthened the justification for using SVD by providing both empirical evidence of its superiority and a theoretical rationale for its necessity.
>
> 1. Empirical Superiority (Sensitivity and Structure):As demonstrated in the comparison with the "Raw Value" baseline in Appendix B.1 and Figure 7, the raw value vectors are highly insensitive to semantic changes.
>
> - Sensitivity: The raw value method requires an extremely high and delicate threshold (e.g., $\tau=0.99$) to detect any separation. In contrast, MindCraft operates robustly with a standard threshold ($\tau=0.9$).
>
> - Structural Clarity: Even with fine-tuning $\tau$, the Raw Value method is not efficient to differentiate semantically distinct concepts effectively, often resulting in degenerate, flattened tree structures where concepts like "diabetes/hypertension" or "2023/2024" remain entangled. MindCraft’s spectral projections successfully resolve these into deep, hierarchical structures.
>
> 2. Theoretical Justification (Principal Component Extraction):The raw weight matrix $W_V$ contains a vast amount of information, much of which may be noise or high-frequency signals irrelevant to the dominant semantic transformation. By applying Singular Value Decomposition (SVD), we extract the principal directions (left singular vectors $U$) associated with the largest singular values.Theoretically, these principal directions represent the axes along which the layer maximally amplifies input signals. By projecting representations onto these axes, we filter out noise and focus on the subspace where the model is most "active" and where semantic concepts are most likely to be encoded. This allows MindCraft to detect subtle conceptual divergences that are buried in the noise of the raw high-dimensional vector space. This also connects with the ablation study of $k$ (Appendix B.2.1), where rural $W_V$ can be seen as a SVD operation with large $k$, which is inefficient to create a hierarchical and meaningful concept tree.
>
> ### W5 & Q2: Multi-Head Attention (MHA) and Internal Representations
>
> Thank you for this question as it allows us to clarify a critical implementation detail regarding how MindCraft handles Multi-Head Attention (MHA).
>
> Definition of Multi-Head Attention: As illustrated in literature [3], MHA is defined as:
>
> $$\text{MultiHead}(Q, K, V) = \text{Concat}(\text{head}_1, \ldots, \text{head}_h) W_O,$$
>
> where $\text{head}_i = \text{Attention}(QW\_{Q(i)}, KW\_{K(i)}, VW\_{V(i)})$, and $W_V = [W\_{V(1)}; W\_{V(2)}; \ldots; W\_{V(h)}]$. By performing SVD on this global $W_V$ (Section 4.2), we use the priliminary $W_V$ projection prior to head partitioning. In other words, we are analyzing the value weight matrix prior to the attention heads at that layer. Therefore we didn't explicitly elaborate MHA in our paper.
>
> Consequently, our method inherently accounts for the complexity of MHA. It treats the layer as a holistic transformation unit, extracting the global spectral directions that drive concept formation, regardless of which specific head attends to them.
>
> [1] Park et al. The Linear Representation Hypothesis and the Geometry of Large Language Models. arXiv preprint arXiv:2311.03658, 2024.
>
> [2] Modell et al. The Origins of Representation Manifolds in Large Language Models. arXiv preprint arXiv:2505.18235, 2025.
>
> [3] Vaswani et al. Attention Is All You Need. NeurIPS, 2017.

---

### Official Review · Reviewer_g9oL · 2025-10-31

**Soundness:** 2
**Presentation:** 2
**Contribution:** 3
**Rating:** 2
**Confidence:** 3

**Summary:**

The paper introduces MindCraft, a framework for analyzing how llm internally structure abstract concepts. The proposed method uses counterfactual input pairs and tracks the difference in the last token's representation across layers. The core of the method which is Concept Path, is computed by projecting the last token's attention value vector onto the principal components of that layer's value transformation matrix. By finding the layer where the Concept Paths of a counterfactual pair first diverge, a hierarchical visualization of when and where different concept split into separable subspaces.

**Strengths:**

1. The paper is well-written and easy to follow.
2. The branch-and-stabilize hypothesis provides a strong, intuitive foundation for the work.
3. The paper demonstrates the flexibility of the proposed framework by applying to multiple model across three domains.

**Weaknesses:**

1. The same citation appears in two different formats.
2. The main text of the figure needs to provide guidance on how to interpret the results in the figure (e.g., what the takeaway is).
3. The paper need to provide a claim that the Concept Tree faithfully represents the model's internal reasoning process.
4. The paper need to clarify the choice of parameters k and tau, or at least analyze the change.
5. The experiment should be compared with baselines (e.g., RepE, LRH). The only baseline comparison is "raw Value" vectors which isn't benchmarked on any metric.

**Questions:**

Look at the weaknesses

---

> ### Author Response · Authors · 2025-11-21
>
> Thank you for your feedback. Below we address raised issues and provide additional clarifications, ablation studies, and baseline comparisons to strengthen the framework.
>
> ### W1. Citations
>
> For citations, we have revised it in the undated version.
>
> ### W2. Clarifying the Interpretation of Figures
>
> We acknowledge that the main text of the paper did not sufficiently guide readers on how to interpret each figure. We have revised the captions and accompanying text for Figures 2–6 to explicitly highlight the *takeaway* from each result:
>
> - **Figure 2** now states that a sudden amplification of the conceptual difference is observed.
> - **Figure 3** now emphasizes the broad applicability of the Concept Tree, revealing a structured and hierarchical organization of conceptual reasoning within the model.
> - **Figure 4** explicitly demonstrate this experiment highlights Concept Tree captures semantic interpretability rather than relying solely on static measures such as input or latent embeddings.
> - **Figure 5** demonstrates the consistent propagation patterns observed across tasks shows that concept formation follows a robust hierarchical dynamic within deep networks.
> - **Figure 6** now highlights the key takeaway that the disentanglement between the branching layer and input embedding distances demonstrates that the Concept Tree reflects high-level conceptual organization beyond what is encoded in embeddings.
>
> And we have indicated the takeaway in each of the new figures we added.
>
> ### W3. The claim that the Concept Tree faithfully represents the model's internal reasoning process
>
> MindCraft is designed as a unifying framework that is built on prior works representing the model's internal reasoning process, that bridges two dominant perspectives on representations in deep models—the **Linear Representation Hypothesis (LRH)** [1] and **the Representation Manifold (RM)** [2].
>
> In the **LRH** view, concepts are encoded as approximately linear directions in the representation space. Formally, the difference between counterfactual activations defines a direction vector:
> $$
> \delta_L = V^{(-1)}_{L(\Delta x)} - V^{(-1)}_L,
> $$
>
> such that $\delta_L$ measures the local linear separability of the two counterfactual concepts. The LRH implies that these concept directions become increasingly linearly separable with depth L, corresponding to the decrease in our Conceptual Separation Score $s_l(X, X_{\Delta x})$ (Eq. 10). The *branching layer* $ l^*$ (Eq. 11) can thus be viewed as the first layer where the representation of a concept transitions from entangled to linearly separable—formally where
> $$
> s_{l}(X, X_{\Delta x}) < \tau.
> $$
>
> This empirically identifies the onset of LRH-style linearization.
>
> Conversely, **RM** posits that internal representations may instead lie on *nonlinear manifolds* $\mathcal{M}_f \subset \mathbb{R}^d$, where distances in representation space encode intrinsic semantic distances between feature values.
> In this setting, the Concept Path $\mathcal{C}_l(X)$ (Eq. 6) trace local trajectories along such manifolds. When concepts only diverge at deeper layers (large $ l^* $), the local curvature of these paths—reflected by gradual rather than abrupt changes in $s_l$—indicates manifold-like unfolding rather than pure linear separation.
>
> MindCraft therefore **unifies these two perspectives** by operationalizing the following interpretation:
>
> - If the concept separation $l^*$ occurs in *early* layers and $s_l$ drops sharply, the concept behaves according to the **linear subspace model** of LRH, implying a nearly constant direction $\delta_L$.
> - If the separation occurs in *late* layers and $s_l$ decreases gradually, the representation follows a **manifold trajectory**, where the concept path $\mathcal{C}_l(X)$ changes smoothly in the principal basis $\\{u_i\\}$.
>
> In summary, **MindCraft** provides an unified observation of how abstract concepts emerge through either *linear separability* (as posited by LRH) and *nonlinear manifold unfolding* (as characterized by RM). This theoretical connection grounds the Concept Tree framework within a broader geometry of representation learning—showing that linear and manifold interpretations are not mutually exclusive but are instead two local regimes of the same underlying representational process. We provide more details in Appendix A.

---

> ### Author Response · Authors · 2025-11-21
>
> ### W4. Parameter Sensitivity ($k$ and $\tau$)
>
> To investigate the effect of the parameter $k$, we conduct an ablation study using the same scenario as in Figure 3 (a), and the result is shown in Figure 8: both extremely small and large values of $k$ cannot effectively capture coherent conceptual structures. When $k$ is too small (e.g., $k = 1$), the model retains only a single dominant spectral direction, losing semantic distinctions. This indicates that conceptual representations within the model are governed by the collective interaction of multiple dimensions rather than by a few isolated ones, suggesting that concept formation is inherently distributed across several principal components.
>
> Conversely, when $k$ is too large (e.g., $k = 100$), excessive noise from less informative components overwhelms the main conceptual signal, resulting in unstable or collapsed Concept Trees. The optimal balance, empirically found at $k = 10$, provides a stable and interpretable hierarchy where concept separations emerge at appropriate layers. This demonstrates that moderate values of $k$ are essential for capturing meaningful conceptual dynamics while maintaining robustness.
>
> We further examine the effect of the separation threshold $\tau$, which determines the layer at which two concepts are separated in the Concept Tree. As illustrated in Figure 9, both overly high and overly low values of $\tau$ fail to yield stable and interpretable structures. When $\tau$ is too high (e.g., $\tau = 0.99$), even minor fluctuations in similarity trigger premature branching, causing the model to over-segment representations and produce shallow and flat trees. In contrast, when $\tau$ is too low (e.g., $\tau = 0.7$), the model delays concept separation until very late layers, also collapsing distinct semantic branches into overly flat trees. The balanced threshold of $\tau = 0.9$ provides the most coherent hierarchy, aligning with the natural emergence of conceptual distinctions observed across layers. This suggests that an appropriate $\tau$ is essential for capturing genuine conceptual divergence without introducing spurious separations.
>
> ### W5. Baseline Comparison and Quantitative Evaluation
>
> *Our work is theoretically orthogonal to Representation Engineering (RepE) [1] and Linear Representation Hypothesis (LRH) [2]*, building a connection between two prevailing yet seemingly contradictory perspectives on representations to advance the theoretical understanding of interpretability.
>
> First, RepE, grounded in the LRH, primarily focuses on extracting linear conceptual directions from latent space. By projecting latent states onto these directions, one can quantify abstract concepts; for instance, by extracting an "honesty/dishonesty" direction, the model can compute an "honesty score" for any input sequence via projection.
>
> However, both RepE and LRH face theoretical limitations when contrasted with the perspective of Representation Manifolds (RM) [3]. This opposing view posits that internal representations are rarely distinct enough to be separated by a simple linear hyperplane. Instead, they often form complex, winding topological structures—for example, the chronological progression of the 20th century manifests as a curved, non-linear manifold within the representation space rather than a linear hyperplane.
>
> The Concept Tree framework overcomes this limitation by providing a unified perspective that bridges the linear view of Representation Engineering with the geometric interpretation of Representation Manifolds. We clarify it in Appendix A.1.
>
> [1] Andy Zou et al. Representation Engineering: A Top-Down Approach to AI Transparency. arXiv preprint arXiv:2310.01405, 2023.
>
> [2] Kiho Park, Yo Joong Choe, and Victor Veitch. The Linear Representation Hypothesis and the Geometry of Large Language Models. arXiv preprint arXiv:2311.03658, 2024.
>
> [3] Alexander Modell, Patrick Rubin-Delanchy, and Nick Whiteley. The Origins of Representation Manifolds in Large Language Models. arXiv preprint arXiv:2505.18235, 2025.

---

### Official Review · Reviewer_yQ4m · 2025-11-05

**Soundness:** 2
**Presentation:** 4
**Contribution:** 3
**Rating:** 6
**Confidence:** 3

**Summary:**

The paper introduces MindCraft, a framework for explaining AI models by constructing concept trees that illustrate the hierarchy of which neural network layers concepts diverge in the internal representations. The tree is constructed by comparing the spectral decompositions of the representations of a counterfactual pair of inputs and splits the tree at the first layer in which the representations begin to differ. Experiments demonstrate clear examples of trees generated for LLM tasks and also highlight some interesting properties of the approach.

**Strengths:**

1. Mindcraft presents a novel framework for interpretable AI that pinpoints precisely where concepts form within the layers of the model.

2. Experiments are comprehensive, easy to understand, and show a variety of properties of the pipeline.

3. The writing is very clear.

**Weaknesses:**

1. There is little theoretical justification of any of the proposed methodology. The final tree is dependent on many parameters left up to the user, and the result is open to vague interpretation. The paper claims that MindCraft “systematically traces how abstract concepts emerge”, but it is unclear why the resulting concept tree answers the “how” question.

2. Counterfactual quantities, from the perspective of causal inference, are not easy to infer, especially in cases where one is attempting to simultaneously infer something about the same input in two different interventions. It is not clear what kinds of assumptions are made to allow for this.

**Questions:**

1. Is there a reason that concept trees are defined in tree format? It seems like all concepts are leaf nodes that branch off of a single main line of nodes that represents the undisambiguated concepts. Could concept trees be instead simply represented as a list of concepts sorted by the order of the layers in which they were disambiguated?

2. Do the patterns that arise in the presented experimental results look similar when applied to non-language tasks?

3. Are the concepts in MindCraft simply defined as sections of the input, or could they represent more abstract concepts produced within the internal workings of the neural network?

---

> ### Author Response · Authors · 2025-11-21
>
> ### W1. Theoretical Justification
>
> MindCraft is designed as a unifying framework that bridges two dominant perspectives on representations in deep models—the **Linear Representation Hypothesis (LRH)** [1] and **the Representation Manifold (RM)** [2].
>
> In the **LRH** view, concepts are encoded as approximately linear directions in the representation space. Formally, the difference between counterfactual activations defines a direction vector:
> $$
> \delta_L = V^{(-1)}_{L(\Delta x)} - V^{(-1)}_L,
> $$
>
> such that $\delta_L$ measures the local linear separability of the two counterfactual concepts. The LRH implies that these concept directions become increasingly linearly separable with depth L, corresponding to the decrease in our Conceptual Separation Score $s_l(X, X_{\Delta x})$ (Eq. 10). The *branching layer* $ l^*$ (Eq. 11) can thus be viewed as the first layer where the representation of a concept transitions from entangled to linearly separable—formally where
> $$
> s_{l}(X, X_{\Delta x}) < \tau.
> $$
>
> This empirically identifies the onset of LRH-style linearization.
>
> Conversely, **RM** posits that internal representations may instead lie on *nonlinear manifolds* $\mathcal{M}_f \subset \mathbb{R}^d$, where distances in representation space encode intrinsic semantic distances between feature values.
> In this setting, the Concept Path $\mathcal{C}_l(X)$ (Eq. 6) trace local trajectories along such manifolds. When concepts only diverge at deeper layers (large $ l^* $), the local curvature of these paths—reflected by gradual rather than abrupt changes in $s_l$—indicates manifold-like unfolding rather than pure linear separation.
>
> MindCraft therefore **unifies these two perspectives** by operationalizing the following interpretation:
>
> - If the concept separation $l^*$ occurs in *early* layers and $s_l$ drops sharply, the concept behaves according to the **linear subspace model** of LRH, implying a nearly constant direction $\delta_L$.
> - If the separation occurs in *late* layers and $s_l$ decreases gradually, the representation follows a **manifold trajectory**, where the concept path $\mathcal{C}_l(X)$ changes smoothly in the principal basis $\\{u_i\\}$.
>
> In summary, **MindCraft** provides an unified observation of how abstract concepts emerge through either *linear separability* (as posited by LRH) and *nonlinear manifold unfolding* (as characterized by RM). This theoretical connection grounds the Concept Tree framework within a broader geometry of representation learning—showing that linear and manifold interpretations are not mutually exclusive but are instead two local regimes of the same underlying representational process. We provide more details in Appendix A.
>
> ### W2. Causal Assumptions
>
> In causal theory, investigating counterfactuals, e.g., in econometrics or clinical trials, is indeed difficult because we cannot observe the outcome of the same unit under two different treatments simultaneously. However, MindCraft operates within the domain of Interpretability of Neural Networks, which differ from observational studies in two ways that relax these constraints:
>
> A neural network is a digital system where we have control over states. We can execute the model with input $X$, record the internal states, and then functionally "reset" the universe to run the exact same model instance with input $X_{\Delta x}$. And because we can run both branches of the counterfactual pair on the exact same parameters, we define the counterfactual difference $\delta Y$ (Eq. 2) as a directly computed vector difference, not a statistically inferred quantity.
>
> These are assumptions that we made:
>
> **Constant Context:** We assume that, tokens other than the intervened token ($X \setminus x$) remain constant.
>
> **Last Token Prediction:** [3,4] We assume that the last-token representation captures the model’s overall generative state and semantic summary of the sequence.
>
> [1] Kiho Park, Yo Joong Choe, and Victor Veitch.
> The Linear Representation Hypothesis and the Geometry of Large Language Models.
> arXiv preprint arXiv:2311.03658, 2024.
>
> [2] Alexander Modell, Patrick Rubin-Delanchy, and Nick Whiteley.
> The Origins of Representation Manifolds in Large Language Models.
> arXiv preprint arXiv:2505.18235, 2025.
>
> [3] Kevin Meng, David Bau, Alex Andonian, and Yonatan Belinkov.
> Locating and Editing Factual Associations in GPT.
> Advances in Neural Information Processing Systems (NeurIPS), 35:17359–17372, 2022.
>
> [4] Andy Zou et al.
> Representation Engineering: A Top-Down Approach to AI Transparency.
> arXiv preprint arXiv:2310.01405, 2023.

---

> ### Author Response · Authors · 2025-11-21
>
> ### Q1: Why a Tree and not a List?
>
> If we were to list concepts sorted by the layer index, we would be making an implicit assumption: that all concepts are independent and measurable on a single, flat metric. First, this view is agnostic to the context; Second, this linear view does not capture the causal dependencies between concepts. In a causal graph, variables are often structured hierarchically, where a "root cause" determines the value or relevance of downstream variables. The Concept Tree visualizes this causal structure.
>
> For instance, in our experimental case (d) on personality evaluation, the concept pair “kind/selfish” separates very early at layer 1, whereas the behavioral pair “help/ignore” separates later at layer 5. From a causal perspective, the fundamental personality trait (kindness) acts as the cause that governs the downstream behavioral manifestation (helping). The model first identifies the core semantic direction—the root concept—which then defines the subspace where specific actions are represented. Conversely, a temporal modifier such as “always/usually” (Layer 9) functions as a causal descendant, modulating the frequency of an already-defined behavior.
>
> A purely linear list would treat “kind/selfish” and “always/usually” as equivalent entities. In contrast, the tree format reveals the model’s decision-making hierarchy: once the branch for kindness is established, it causally enables the subsequent differentiation of helping. This structure exposes the model’s layered reasoning process, beyond depicting a sequential measurement of token processing.
>
> ### Q2: Applicability to Non-Language Tasks
>
> Yes, the patterns and methodology are expected to generalize because MindCraft operates on the Attention mechanism, not language-specific heuristics.The core of MindCraft is the spectral decomposition of the Value ($V$) projection and the analysis of attention dynamics. Since our method calculates the propagation of differences through the attention mechanism (Eq. 3-4), it is theoretically applicable to any architecture that employs the Transformer backbone. Current SOTA models in various domains, such as Vision-Language Models (VLMs [1]) or Vision Transformers (e.g., DinoV2 [2]), utilize the same fundamental attention structures.
>
> ### Q3: Input Sections vs. Abstract Concepts
>
> Quick answer: We use counterfactuals of input sections to represent abstract concepts.
>
> Details: The concepts is defined through counterfactuals, which is based on the intervention and context of the input tokens. While the intervention targets a specific input token, the Concept Tree captures the counterfactual difference between representations. Specifically, a counterfactual [3] is defined as $Y_{x \leftarrow \Delta x} =Y(do(x = \Delta x) , X \setminus x \bigr)$. By keeping the context identical and only intervening on the target $x$, we are forcing the model to process intervention: "What if the meaning of this entity were different, all else being equal?" This isolates the conceptual difference from background noise. Therefore, MindCraft visualizes the emergence of the abstract concepts.
>
> [1] Jiasen Lu, Dhruv Batra, Devi Parikh, and Stefan Lee. ViLBERT: Pretraining Task-Agnostic Visiolinguistic Representations for Vision-and-Language Tasks. Advances in Neural Information Processing Systems (NeurIPS), 32, 2019.
>
> [2] Maxime Oquab, Timothée Darcet, Theo Moutakanni, Huy Vo, Marc Szafraniec, Vasil Khalidov, Pierre Fernandez, Daniel Haziza, Francisco Massa, Alaaeldin El-Nouby, et al. DINOv2: Learning Robust Visual Features without Supervision. arXiv preprint arXiv:2304.07193, 2023.
>
> [3] Judea Pearl. Causality: Models, Reasoning, and Inference. Cambridge University Press, 2nd edition, 2009.

---

### Meta-Review · Area_Chair_CWDr · 2026-01-06

**Summary:**

The main concerns of the reviewers are:

Lacks a theoretical justification of any of the proposed methodologies.

It is not clear what kinds of assumptions are made to allow for the counterfactual quantities.

Clarity of the figures, parameters.

Wrong citations.

Lacks enough baselines and models for testing.

The results presented in the experimental section are mostly qualitative.

**Reviewer Concerns:**

The concerns about the lack of theoretical justifications, enough experiments including more baselines and base models, and modalities remain. And some of the assumptions also need more detailed discussion to provide enough context for the readers.

**Reviewer Scores:**

I think the main critical reviewers would not change the negative score. The other two reviewers mainly hold their ratings.​

---

### Decision · Program_Chairs · 2026-01-26

Reject